# Towards Knowledge-and-Data-Driven Organic Reaction Prediction: RAG-Enhanced and Reasoning-Powered Hybrid System with LLMs

**Qingyu Wang**[1,2*] **Xinyuan Cai**[1*] **Xiang Cheng**[1] **Yuzhe Gao**[1,2] **Bo Xu**[1†]

[1] The Key Laboratory of Cognition and Decision Intelligence for Complex Systems,
Institute of Automation, Chinese Academy of Sciences
[2] School of Artificial Intelligence, University of Chinese Academy of Sciences, Beijing, China

## Abstract

In organic reaction prediction, many recent approaches ranging from traditional task-specific models to Large Language Models (LLMs), have demonstrated notable success. However, these methods are inherently data-driven, exhibit constrained interpretability, and have hit fundamental performance bottlenecks. To overcome these limitations, we present **Reaction-Thinker**, a hybrid, knowledge-and-data-driven system that is enhanced by Retrieval-Augmented Generation (RAG) and powered by advanced reasoning, improving both the interpretability of prediction process and the explainability of results. We develop a similar-case retrieval database and train a RAG-based LLM through supervised fine-tuning (SFT) to apply both reaction types and similar reaction cases as knowledge. We also construct a reaction reasoning chain-of-thought (CoT) dataset and train a reasoning-based LLM through SFT, then further optimize it using Group Relative Policy Optimization (GRPO). Experimental results show that our method outperforms all compared LLMs and task-specific models, achieving the highest accuracy (Exact Match) and fingerprint similarity (FTS). Ablation study indicates improvements in relative accuracy of 7.5% and 13.9% for RAG and GRPO, respectively. Further analysis of mispredictions reveals limitations in conventional evaluation metrics, which motivates our proposed benchmarking refinement.

## 1 Introduction

In organic chemistry, predicting reaction outcomes has long been a core challenge. Traditionally, expert chemists relied on years of hands-on experience and well-established principles to design experiments and anticipate products. Today, artificial intelligence offers a powerful augment to this approach, enhancing the efficiency and precision of prediction.

Current approaches to predicting organic reaction outcomes can be mainly categorized into template-based and template-free methods. Template-based methods integrate machine learning with predefined structural transformation rules, also known as reaction templates, either curated by experts or extracted from atom-mapped datasets (Chen & Jung, 2022; Sacha et al., 2021). In contrast, template-free methods employ data-driven architectures, such as graph neural networks (GNNs) or Transformer-based sequence models, to infer reaction patterns directly from large reaction corpora without relying on explicit templates. (Schwaller et al., 2018; Irwin et al., 2022). Recent advances in large language models (LLMs) have garnered significant attention (Achiam et al., 2023; Team et al., 2024; Bai et al., 2023; Liu et al., 2024). By undergoing large-scale pre-training followed by fine-tuning or instruction tuning, these models acquire extensive knowledge, proficiently follow instructions, and exhibit strong reasoning abilities. As a result, LLMs now achieve state-of-the-art

---

*These authors contributed equally.

†Corresponding author. Email: xubo@ia.ac.cn

(SOTA) performance comparable to or even exceeding that of humans, in general Natural Language Processing (NLP) tasks such as language understanding and question answering, as well as specialized applications including mathematical problem-solving and code generation. Hence, a natural idea is to explore whether LLMs can replicate the cognitive processes of expert chemists, enabling more accurate reaction predictions.

Human chemists predict organic reactions through a multi-step cognitive process. Initially, they analyze molecular structures, identifying functional groups, bond connectivity, stereochemistry, and reactive sites, which are fundamental to mechanistic analysis (Smith, 2023). Then, they apply core principles to hypothesize bond cleavage and formation. They propose reaction pathways and elucidate mechanistic steps, including identifying reaction centers, considering mechanisms such as SN1, SN2, or pericyclic, and evaluating thermodynamic and kinetic feasibility (Levy, 2017). Finally, they integrate insights to predict the main product and account for side reactions. Apart from that, known reaction cases are also frequently referenced to inform predictions.

Recent works have applied LLMs to chemistry, particularly targeting organic reactions prediction. Notable examples include the ChemDFM series (Zhao et al., 2024b;a), ChemLLM (Zhang et al., 2024), ChemCrow (M. Bran et al., 2024), and Coscientist (Boiko et al., 2023). Some approaches leverage large proprietary models such as GPT-4o (OpenAI, 2024) directly, exploiting their innate zero-shot reasoning capabilities. Others build on open-source LLMs such as LLaMA (Touvron et al., 2023) and fine-tune them on chemical literature and curated datasets, resulting in domain-specific models with enhanced accuracy on chemical question answering (Q&A) and prediction tasks.

However, current fine-tuning methods for LLMs in chemistry primarily rely on data-driven strategies, which are similar to traditional end-to-end deep learning approaches, and often fail to fully leverage the rich chemical knowledge embedded in the pre-trained parameters of LLMs, and finetuned models tend to underutilize their robust reasoning and in-context learning capabilities. Consequently, the predictions often lack interpretability and do not outperform established task-specific methods in accuracy. Although LLMs hold immense promise for organic reaction prediction, owing to their pre-trained chemical knowledge as well as robust in-context learning and reasoning capabilities, two critical bottlenecks must still be addressed before this potential can be fully realized.

First, high-quality, structured training data is severely scarce in chemistry. Domains like mathematics benefit from extensive open-source communities (e.g., Lean Community) and web-scale datasets, whereas chemistry lacks publicly available resources for reaction reasoning. As a result, models undergo pre-training and fine-tuning with limited exposure to task-relevant chemical data, hindering their ability to develop advanced capabilities for complex reaction prediction tasks. Furthermore, creating large-scale, annotated chemical datasets is a time-consuming and labor-intensive process that demands substantial domain expertise.

Second, learning strategies for chemistry LLMs remain underdeveloped. Most existing chemical LLM frameworks rely on standard pre-training followed by supervised fine-tuning (SFT), which often fails to unlock the full potential of these models. Recent research highlights two advanced techniques, including Retrieval-Augmented Generation (RAG) (Ke et al., 2024; Chen et al., 2025), which can inject domain-specific knowledge as well as mitigating hallucinations, and reinforcement learning (RL) (Guo et al., 2025), which can enhance reasoning and interpretability. However, their adoption in chemical LLMs remains limited. Therefore, developing a hybrid learning framework that integrates SFT, RAG and RL, combining both data-driven and knowledge-driven paradigms, represents a promising direction for achieving interpretable, high-performance prediction.

In this paper, we propose **Reaction-Thinker**, a hybrid knowledge-and-data-driven organic reaction prediction system, comprising both a RAG-based predictor and a reasoning-based predictor. The main contributions of our work can be concluded in the following parts.

- We categorize reactions based on the given reaction inputs, define a standardized similarity metric, and construct similar-case retrieval database for each type. Training and test samples are partitioned based on whether similar retrieved cases exist, and each subset is processed through a dedicated, specialized pipeline.

- For samples with similar cases retrieved, we inject reaction type and case-specific knowledge into user prompt and curate a customized SFT dataset for a RAG-based LLM. This enhances the ability of the LLM to retrieve domain-specific contextual information.

- For samples lacking similar reaction cases, we introduce a multi-stage reasoning enhancement pipeline. First, we construct a chain-of-thought (CoT) dataset for organic reaction reasoning. Then, we employ SFT as a cold-start to establish an initial foundation of high-accuracy CoT reasoning. Finally, we refine the deductive reasoning using reinforcement learning through Group Relative Policy Optimization (GRPO).

- The system outperforms all compared LLMs and even exceeds traditional task-specific models, in both accuracy (Exact Match) and fingerprint similarity performance (FTS). The ablation study indicates improvements in relative accuracy of 7.5% and 13.9% for RAG and GRPO, respectively.

- A detailed error analysis reveals that some incorrect predictions correspond to chemically plausible byproducts or alternative reaction pathways, despite not matching the canonical ground truth. To better account for such chemically plausible outcomes, we propose a novel evaluation paradigm by incorporating retrosynthetic validation. Notably, our analysis indicates that 47.8% of these incorrect predictions are chemically reasonable.

## 2 METHODS

As illustrated in Figure 1, our proposed organic reaction predict system integrates four core modules: (1) a reaction type classifier, (2) a similar-case retrieval database, (3) a RAG-based reaction predictor, and (4) a reasoning-based reaction predictor. Given a set of reaction inputs including reactants, solvent, and reagents, the system first applies the classifier to determine the most probable reaction type. Based on the predicted type, it queries the similar-case retrieval database for analogous reaction examples. If similar reaction cases are found, they are then incorporated alongside the user prompt into the RAG-based predictor; otherwise, the reaction inputs are routed directly to the reasoning-based reaction predictor for CoT-based analysis. Depending on the pathway, either RAG-enhanced or reasoning-based module generates the final reaction outcome.

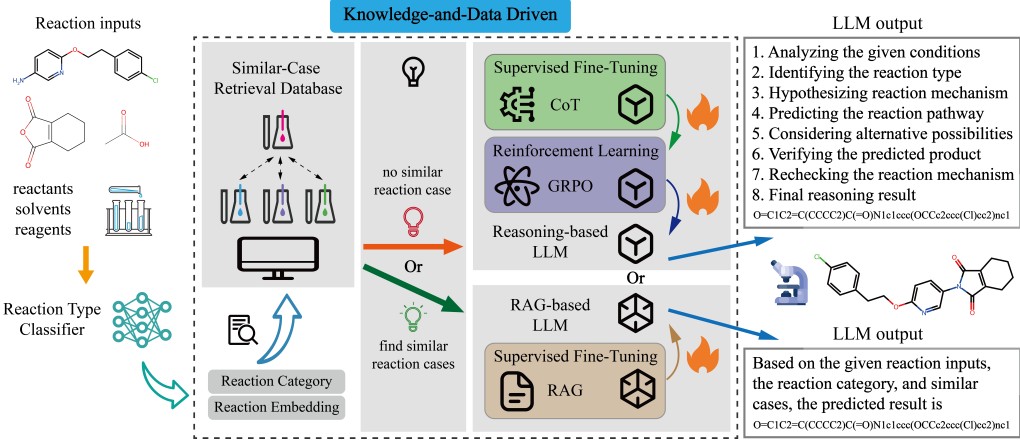

Figure 1: The system architecture, training process, and inference pipeline of **Reaction-Thinker**.

Subsequent sections detail the training process of reaction type classifier, the construction and usage of the similar-case retrieval database, the preparation of a CoT dataset for organic reaction reasoning, and the training strategies for both RAG-based and reasoning-based predictors.

### 2.1 REACTION TYPE CLASSIFIER

We implement a two-layer MLP as the classifier. The original reaction inputs, provided as SMILES strings, are processed with RDKit (Landrum, 2016) to compute multiple structural fingerprints. These fingerprints ($Mol\text{-}Fingerprint$) are then concatenated and fed into the MLP to predict reaction type ($Classifier\text{-}Out$).

Inspired by previous work (Safizadeh et al., 2021), we employ a combination of various molecular fingerprint methods to comprehensively capture molecular information, including RDK (suitable for general molecular similarity searches) (Schneider et al., 2015b), LAYERED (useful for substructure screening) (RDKit-Book, 2025), PATTERN (focused on identifying specific chemical features) (RDKit-Book, 2025), AVALON (effective for both substructure screening and similarity matching in complex molecules) (Gedeck et al., 2006), and MORGAN fingerprints (especially suitable for cyclic substructure and comparing structural features) (Rogers & Hahn, 2010).

We train the reaction type classifier on the Schneider-50K dataset (Schneider et al., 2015a), which contains 50K reaction SMILES classified into 50 representative types, providing granularity well-suited for robust classification. After training, we extract the first layer output of the classifier as a compact representation of the reaction inputs, providing a molecular embedding ($Rea\text{-}Embedding$) of the reaction. The architecture of the reaction type classifier is described in Equation (1):

$$
\begin{aligned}
Rea\text{-}Embedding &= \boldsymbol{Layer1}\left(Mol\text{-}Fingerprint\right) \\
Classifier\text{-}Out &= \boldsymbol{Layer2}\left(Rea\text{-}Embedding\right)
\end{aligned}
\tag{1}
$$

## 2.2 SIMILAR-CASE RETRIEVAL DATABASE FOR RAG

For each reaction in both training and test splits of Open Reaction Dataset (ORD) (Kearnes et al., 2021), we feed its reaction inputs (including reactants, solvents, and reagents, all formatted as SMILES) into our trained reaction type classifier. This yields two outputs: (1) a molecular embedding from the first layer, and (2) a predicted reaction type.

Using the ORD training set, we construct a similar-case retrieval library for each reaction type by computing the Euclidean distance ($L_2$ norm) between the molecular embedding of each query reaction and that of all other embeddings of the same type. Any training sample with an embedding distance smaller than $M$ is considered similar, and its full reaction SMILES (including reactants, solvents, reagents, and products) will be added to the retrieval library for that type.

Based on this, we build a SFT dataset for RAG-based LLM, containing only reactions that successfully retrieve at least one similar case. Each SFT sample includes the reaction inputs, its predicted reaction type, the retrieved similar cases, and the target products.

During inference, we apply the same embedding and classification process to reaction inputs from the test set. If the system retrieves one or more training cases within distance $M$, it follows the RAG pathway; otherwise, it proceeds to the reasoning pathway without external context.

## 2.3 CoT DATASET FOR REACTION REASONING

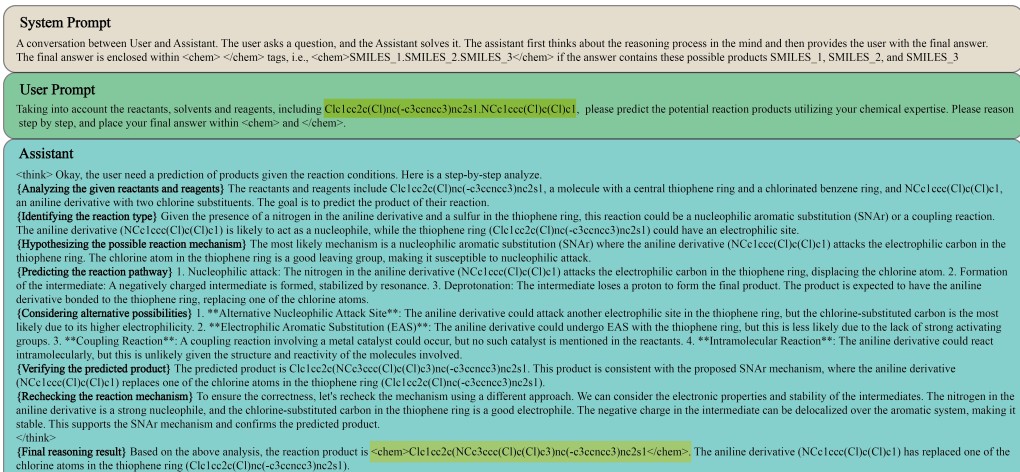

Figure 2: An example of chain-of-thought dataset for reasoning, including system prompt, user prompt, and supervised output.

Here a two-stage approach is adopted to generate CoT data for reaction reasoning, involving both the USPTO-MIT (Jin et al., 2017) and ORD. Finally, we merge CoT samples obtained from both stages into a unified dataset, serving as the primary CoT resource for this work. The data are open-sourced (refer to link and details in **Appendix**).

### 2.3.1 STAGE 1: PRELIMINARY CONSTRUCTION

Following the methodology in HK-O1aw (Lab, 2024), we extract reaction SMILES from a random subset of the USPTO-MIT training set. We have carefully filtered the USPTO-MIT training data to ensure that no reaction SMILES from it appears in the ORD test set, eliminating any risk of data leakage during training and inference. These SMILES were processed using Qwen2.5-72B-Instruct (Qwen Team, 2024) with instructions to reconstruct the reaction mechanism through deductive reasoning, systematically deriving the products from the given reactants, solvents, and reagents through a chain-of-thought reasoning process, even though the model has access to the final answer (refer to **Appendix** for the prompt details). We then apply rigorous post-processing to refine the generated contexts, including format standardization and keyword-based validation, ultimately obtaining 119K high-quality CoT samples after careful filtering. An example of the dataset is shown in Figure 2. Although directly predicting reaction outcomes from conditions is challenging, our approach leverages the observation that when provided with full reaction SMILES, LLMs can systematically deduce the reaction pathway by analyzing the transformation from reactants to products.

### 2.3.2 STAGE 2: DISTILLATION AND VALIDATION

We first fine-tune DeepSeek-R1-Distill-Qwen-7B (DeepSeek-AI, 2025c) on the CoT samples generated from USPTO-MIT using SFT. Then, we further train the model on the ORD training set using Group Relative Policy Optimization (GRPO) (see experimental results in **Appendix**). The GRPO will be explained in detail in the following section. During this stage, only those generated reasoning trajectories that lead to correct predicted products are retained. Overall, we collected 575K validated CoT examples, covering approximately 55K unique samples from the original ORD dataset.

## 2.4 TRAINING STRATEGY OF RAG-BASED LLM

### 2.4.1 SUPERVISED FINE-TUNING

We fine-tune RAG-based LLM on the dataset constructed from similar-case retrieval database, using SFT with full parameter updates. The Qwen3-32B (Qwen Team, 2025) is selected as the backbone model for this process.

## 2.5 TRAINING STRATEGY OF REASONING-BASED LLM

We employ a two-stage training strategy (SFT followed by RL) for our reasoning-based LLM, using DeepSeek-R1-Distill-Qwen-32B (DeepSeek-AI, 2025b) as the backbone model due to its strong reasoning performance compared to other LLMs of similar size.

### 2.5.1 SUPERVISED FINE-TUNING

First, we fine-tune the base model using SFT with full parameter updates on the generated CoT dataset for reaction reasoning. Through this process, the model begins to internalize reasoning patterns specific to organic reaction.

### 2.5.2 REINFORCEMENT LEARNING

Next, we perform RL with LoRA adapters (Hu et al., 2022) on the ORD training set to further enhance reasoning accuracy as well as reliability.

Specifically, we use GRPO as the learning algorithm. Given an input query $q \sim P(Q)$, GRPO samples a group of $G$ responses $\{y_1, y_2, \ldots, y_G\}$ from the current policy $\pi_{\theta_{old}}$. The core idea is to update the policy $\pi_\theta$ by maximizing an objective function that encourages responses with higher-

than-average rewards within their group. The GRPO objective function is defined as follows:

$$\mathcal{J}(\theta) = \mathbb{E}_{q \sim P(Q),\ \{y_i\}_{i=1}^{G} \sim \pi_{\theta_{old}}(\cdot | q)} \left[ \frac{1}{G} \sum_{i=1}^{G} \frac{1}{|y_i|} \sum_{t=1}^{|y_i|} \right.$$

$$\left. \min \left( c_{i,t}(\theta) \hat{A}_{i,t}, \text{clip}\left( c_{i,t}(\theta), 1 - \epsilon, 1 + \epsilon \right) \hat{A}_{i,t} \right) - \beta\, \mathbb{D}_{KL}\left[ \pi_\theta \,\|\, \pi_{ref} \right] \right] \tag{2}$$

where $\epsilon$ is the clipping ratio, $\beta$ is the coefficient for KL-divergence loss, and $\pi_{ref}$ is the reference policy. $c_{i,t}(\theta)$ is the importance sampling ratio for token $y_{i,t}$ (the $t$-th token of the $i$-th response $y_i$):

$$c_{i,t}(\theta) = \frac{\pi_\theta(y_{i,t}|q, y_{i,<t})}{\pi_{\theta_{old}}(y_{i,t}|q, y_{i,<t})} \tag{3}$$

$\hat{A}_{i,t}$ is advantage estimate for all tokens in response $y_i$ and is calculated by normalizing the rewards $\{r_1, r_2, \ldots, r_G\}$ using the group mean and standard deviation:

$$\hat{A}_{i,t} = \frac{r_i - \text{mean}(\{r_1, r_2, \ldots, r_G\})}{\text{std}(\{r_1, r_2, \ldots, r_G\})} \tag{4}$$

### 2.5.3 REWARD FUNCTIONS

Here we design a custom reward function for RL, specifically tailored to organic reaction reasoning, composed of four components:

- **Format Reward:** Assess whether the response format strictly follows the user instructions, awarding 0.1 for correct compliance; or otherwise zero.
- **Length Reward:** Encourages concise reasoning, awarding 0.1 if the chain-of-thought length falls within a predefined range (500 to 2000 tokens); or otherwise zero.
- **Validity Reward:** Assess the validity of the generated product SMILES, awarding 0.1 for chemically valid; or otherwise zero.
- **Accuracy Reward:** Canonicalize the SMILES of generated product and compare it to the ground truth. Award 2.0 if the two match exactly; or otherwise zero.

The final reward is calculated as the sum of these components. This composite reward structure ensures that the model is incentivized to produce well-formed, appropriately concise, and chemically accurate reasoning.

## 3 EXPERIMENTS

In this section, we first present a comprehensive comparative evaluation between our method and existing baselines. Next, we evaluate the contribution of RAG. We also conduct an ablation study to assess how variations in reward function design and the use of cold-start strategy affect GRPO performance in this task. Finally, through detailed analysis of mispredictions, we identify critical limitations in conventional evaluation metrics. Leveraging these insights, we propose a novel reference metric and present its evaluation result. Additional details are provided in **Appendix**.

### 3.1 EXPERIMENTAL SETTING

#### 3.1.1 DATASET AND METRICS

The raw Open Reaction Database (ORD) has been preprocessed by ORDerly (Wigh et al., 2024) and split into 832K for training and 86K for testing. For evaluation, we employ various metrics including Validity (whether the product SMILES can be successfully processed by RDKit), Exact Match (after canonicalization) and molecular fingerprint similarity (FTS, including MORGAN, RDK, and AVALON fingerprints). We deliberately avoid relying on text-based similarity metrics (e.g., BLEU and Levenshtein), since they poorly reflect actual molecular differences, even a single alteration in a SMILES string can correspond to a substantial change in the chemical structure.

### 3.1.2 BASELINES

We compare our system with: (1) open-source LLMs including GPT-4o, DeepSeek-R1 (Guo et al., 2025), DeepSeek-R1-Distill-Llama-70B (DeepSeek-AI, 2025a), DeepSeek-R1-Distill-Qwen series (32B/14B/7B), and Qwen2.5-72B-Instruct; (2) chemical LLMs including ChemDFM-13B/8B (OpenDFM Team, 2024) and Text-Chem-T5 (Christofidellis et al., 2023); and (3) traditional task-specific models including Chemformer, which is reported to achieve the best Top-1 accuracy on USPTO-MIT (Chen & Jung, 2022), and Molecular Transformer (Schwaller et al., 2019).

For open-source LLMs, we use the same user prompt template as our method, and for chemical LLMs, we adopt the training prompts specified in the relevant papers.

## 3.2 MAIN RESULT

As shown in Table 1, our method outperforms all compared LLMs and traditional task-specific models, achieving the highest Exact Match and fingerprint similarity (FTS).

Table 1: Comparison of our method with various baselines on the task of organic reaction prediction. The results for Molecular Transformer are directly taken from existing work (Wigh et al., 2024). Due to API cost constraints, the results for GPT-4o (marked †) are based on a randomly sampled subset of 500 instances from the ORD test set. The top results are highlighted in **bold**.

| Model | Model Type | Validity (%) | Exact Match (%) | FTS (%) | | |
| --- | --- | --- | --- | --- | --- | --- |
| | | | | MORGAN | RDK | AVALON |
| Chemformer | Task-Specific Model | 98.57 | 88.13 | 92.40 | 94.35 | 95.12 |
| Molecular Transformer | | **99.66** | 85.84 | – | – | – |
| GPT-4o † | General LLM | 83.05 † | 28.26 † | 64.93 † | 72.13 † | 72.09 † |
| DeepSeek-R1 | | 68.54 | 11.68 | 55.71 | 64.09 | 64.98 |
| Qwen2.5-72B-Instruct | | 35.27 | 0.54 | 37.46 | 45.95 | 46.01 |
| DeepSeek-R1-Distill-Llama-70B | | 66.42 | 7.20 | 49.06 | 59.17 | 59.67 |
| DeepSeek-R1-Distill-Qwen-32B | | 57.58 | 6.52 | 50.50 | 61.05 | 61.05 |
| DeepSeek-R1-Distill-Qwen-14B | | 43.72 | 1.69 | 32.72 | 40.92 | 40.95 |
| DeepSeek-R1-Distill-Qwen-7B | | 47.74 | 1.21 | 43.50 | 54.58 | 53.77 |
| ChemDFM-13B | Chemical LLM | 98.29 | 52.41 | 77.27 | 82.03 | 82.15 |
| ChemDFM-8B | | 97.80 | 48.02 | 74.69 | 79.85 | 80.02 |
| Text-Chem-T5 | | 95.67 | 47.88 | 76.45 | 81.81 | 81.81 |
| **Reaction-Thinker (Ours)** | – | 98.92 | **89.86** | **95.22** | **96.24** | **96.37** |

The final performance of our method is achieved through integration. In the test set, 81.7% of the samples have similar reaction cases available. For these, our RAG-based approach achieves an Exact Match of 94.70%. For the remaining 18.3% samples without similar cases, we apply reasoning-based approach and achieve an Exact Match of 68.24%. Combining these two approaches yields an overall accuracy of 89.86% across the entire test set. The FTS score is computed using the same weighted approach, combining the RAG-based and reasoning-based results according to their respective proportions.

## 3.3 ABLATION STUDY

### 3.3.1 CONTRIBUTION OF RAG

The effectiveness of RAG has been widely documented in recent works. By grounding generation with retrieved context, it significantly reduces hallucinations and improves accuracy across many domains. To verify the benefit of RAG for current task, we continue to use Qwen3-32B as the base model and conduct a controlled comparison. Rather than following the conventional RAG setup that retrieves reaction types and similar cases for prompt augmentation, we perform a direct end-to-end supervised fine-tuning, mapping reaction input SMILES directly to product SMILES. As shown in Table 2, using RAG yields better performance, with a relative accuracy improvement of 7.5%. This matches chemical intuition: just as chemists reference analogous reactions, LLMs benefit from RAG to improve prediction accuracy.

Table 2: Accuracy performance with and without RAG.

|  | w/ RAG | w/o RAG (End-to-End) |
|---|---|---|
| Exact Match (%) | **83.13** | 77.35 |

### 3.3.2 RAG ABLATION STUDY

We also explore how the embedding distance threshold $M$ (which determines what reaction case counts as similar, and is detailed in Section **Similar-Case Retrieval Database for RAG**) affects both the proportion of queries having similar cases and the RAG pathway performance (in Table 3).

Table 3: Exact Match performance under different RAG thresholds.

| $M$ | Proportion with similar cases (%) | RAG-based Exact Match (%) | Reasoning-based Exact Match (%) | Total Acc (%) |
|---|---|---|---|---|
| 10 | 81.70 | 94.70 | 68.24 | **89.86** |
| 30 | 92.52 | 87.33 | 69.78 | 86.02 |
| 40 | 94.68 | 87.99 | 68.15 | 86.93 |
| 100 | 99.10 | 88.94 | 67.27 | 88.74 |

The results show that tight thresholds (e.g. $M$=10) yield high Exact Match (94.70%) but for fewer queries, while looser thresholds increase coverage yet degrade accuracy. This highlights the need to identify an optimal operating point. Ultimately, we selected $M$=10 in this work because it yields the highest overall accuracy (Total Acc).

### 3.3.3 INFLUENCE OF GRPO

GRPO is an effective reinforcement learning framework for LLMs, where the design of reward functions and selection of the base model critically determine its performance. To further explore its application in current task, we conduct two controlled experiments.

**1. Reward Function Ablation Study**

Building on the baseline reward function (comprising format, length, validity, and accuracy rewards), we introduce a MORGAN fingerprint similarity reward (FTS reward, ranging from 0.0 to 1.0) in order to structurally align predictions with ground truths.

This modification aims to mitigate reward sparsity by guiding the LLM to generate outputs from structurally similar to fully accurate. We record the reward curves during training in Figure 3 and evaluate the resulting model with results presented in Table 4. The experiments reveal a paradoxical phenomenon, while the reward curve shows continuous improvement, the evaluation accuracy actually declines. Through detailed analysis, we identify that the fine-tuned LLM tended to verbatim copy reactant SMILES in outputs. This indicates a suboptimal optimization strategy, since product and reactants structures share chemical similarities, directly copying reactants could still achieve relatively high reward.

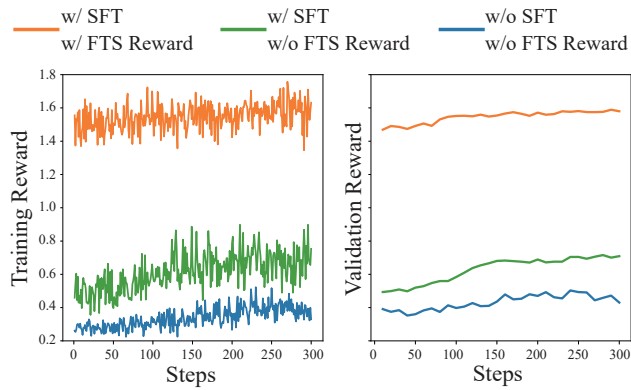

Figure 3: The reward curves under different combinations of (i) whether SFT was applied before RL and (ii) whether the FTS reward is introduced.

Table 4: Accuracy performance with and without fingerprint similarity reward in GRPO.

|  | w/ FTS reward | w/o FTS reward |
| --- | --- | --- |
| Exact Match (%) | 56.83 | **68.24** |

To address this issue, we downweight the FTS reward and incorporate explicit penalties for reactant copying. However, subsequent experiments demonstrate these measures are insufficient to completely prevent this behavior. This phenomenon exemplifies reward hacking, a well-documented RL failure mode where agents optimize the proxy reward in unintended ways, achieving higher scores while failing the true task objective. The tendency of LLM to cheat by exploiting structural correlations between reactants and products remains a significant challenge, now recognized as a critical focus for our ongoing optimization efforts.

**2. Base Model Capability Analysis**

To evaluate the impact of base model capability on GRPO performance, we conduct a controlled experiment comparing two approaches: (1) direct application of GRPO to the initial DeepSeek-R1-Distill-Qwen-32B, and (2) implementing GRPO following SFT on our reaction reasoning dataset.

The experimental results, including the reward curves in Figure 3 and final performance in Table 5, reveal markedly different outcomes between the two settings. This underscores that proper initialization through SFT is critical for unlocking the potential of GRPO in reaction reasoning tasks.

Table 5: Exact Match performance (%) under different SFT and GRPO settings. On the base model with enhanced initial reasoning capabilities (w/ SFT), applying GRPO yields a relative accuracy improvement of 13.9%.

|  | w/o SFT | w/ SFT |
| --- | --- | --- |
| w/o GRPO | 6.52 | 59.93 |
| w/ GRPO | 9.67 | **68.24** |

## 3.4 ANALYSIS OF INCORRECT PREDICTIONS

Through systematic error analysis comparing LLM predictions with ground truth, we identified two major failure modes: (1) complex reactions involving multiple functional groups or multi-step processes, and (2) incomplete reaction conditions (e.g., missing temperature or catalysts, which is confirmed by GPT-4o). Figure 4 presents an example of incorrect prediction along with detailed analysis using GPT-4o and retrosynthetic validation.

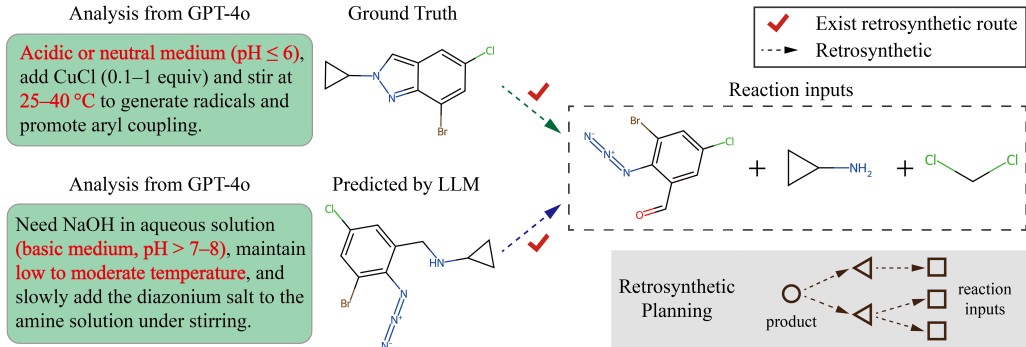

Figure 4: Analysis of incorrect prediction using GPT-4o and retrosynthetic validation. The key reaction conditions influencing final product are marked in bold red. Both products are valid candidate answers in the absence of specific condition constraints.

Fundamentally, many organic reactions inherently generate byproducts via parallel or competing pathways, yet existing datasets typically record only one to three major products. This exposes a

critical limitation in current evaluation metrics for reaction prediction tasks, where exclusive comparison to a single ground truth fails to reflect chemical reality and may hinder LLMs from developing a genuine understanding of organic reaction mechanisms.

To address this, we propose a novel evaluation paradigm by incorporating retrosynthetic validation. For each product predicted by reasoning-based LLM, we verify whether a plausible retrosynthetic route exists based on the given reaction inputs. If chemically reasonable, a prediction is deemed correct even if it does not match the ground truth. Applying the retrosynthetic analysis tool Retro* (Chen et al., 2020) to our previously mispredicted examples, 47.8% of them are validated as chemically reasonable, bringing the total fraction of reactions passing retrosynthesis validation to 92.64%. **Appendix** reports experiments for the baseline models using the retrosynthetic validation method.

## 4 CONCLUSION

In this study, we introduce **Reaction-Thinker**, a hybrid, knowledge-and-data-driven system that significantly advances organic reaction prediction by combining RAG-based LLM with reasoning-based LLM. Experiments on ORD demonstrate that our system achieves state-of-the-art (SOTA) results in Exact Match and fingerprint similarity, outperforming all compared LLMs and task-specific models. The result shows the potential of leveraging LLMs for addressing fundamental challenges in chemical research. We also identify several promising directions for future enhancement. For example, the reasoning-based LLM still has significant room for improvement. The optimization of reward functions will be our focus in further research to help LLMs have better understanding of organic reaction mechanisms. Moreover, enhanced CoT datasets incorporating chemical synthesis processes will be developed and integrated into current training framework, enabling more rigorous analysis of how reaction condition variations affect outcomes. Last but not least, current system implements RAG and reasoning as separate LLM modules, and future work will integrate these capabilities into a unified architecture.

## REPRODUCIBILITY STATEMENT

We have made the code and data publicly available to support reproducibility. The experimental details and GitHub link can be found in **Appendix**.

## ACKNOWLEDGMENTS

This work was supported by the Strategic Priority Research Program of the Chinese Academy of Sciences under Grant No. XDA0480303 and the National Science and Technology Major Project under Grant No. 2023ZD0120901. The AI-driven experiments, simulations, and model training were performed on the robotic AI-Scientist platform of the Chinese Academy of Sciences.

## ETHICS STATEMENT

This research complies with the ICLR Code of Ethics. Our study uses publicly available benchmark data and does not involve human subjects or collection of sensitive information. The authors declare no potential conflicts of interest or sponsorship that could influence the work reported in this paper.

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

# A APPENDIX

## A.1 RELATED WORK

### A.1.1 RETRIEVAL-AUGMENTED GENERATION IN LLMS

RAG can be an effective paradigm for infusing LLMs with non-parametric knowledge (Gao et al., 2023; Gupta et al., 2024; Li et al., 2025), with demonstrated impact in knowledge-intensive domains such as medicine (Li et al., 2024) and law (Wiratunga et al., 2024). By retrieving and conditioning on external documents, RAG significantly improves performance on generation tasks. RAG methods can be broadly categorized into single-round and multi-round strategies. Basic RAG approaches typically retrieves knowledge based solely on the initial query (Guu et al., 2020; Borgeaud et al., 2022). Some works have also explored multi-round retrieval strategies that iteratively refine or rewrite queries across steps (Shao et al., 2023; Jiang et al., 2023), interleave retrieval with reasoning (Trivedi et al., 2022), or utilize multi-stage self-asking mechanisms (Press et al., 2022). Depending on the task, either single-round or multi-round retrieval strategies can be employed.

### A.1.2 REINFORCEMENT LEARNING FOR COT REASONING

Chain-of-Thought (CoT) reasoning (Trivedi et al., 2022) represents a significant methodological advancement in enhancing the reasoning capabilities of LLMs. This approach prompts models to explicitly generate intermediate reasoning steps before arriving at a final output. Such structured reasoning processes substantially improve prediction accuracy, as higher-quality intermediate contexts often contribute to more reliable and consistent final results. Reinforcement learning (RL) has also emerged as a powerful technique for improving the reasoning ability of LLMs, particularly in domains such as mathematics, where structured reward signals allow models to learn beyond what SFT alone can achieve. Recent developments have introduced RL frameworks with numerical feedback, often relying on online policy optimization algorithms such as Proximal Policy Optimization (Schulman et al., 2017), Group Relative Policy Optimization (GRPO) (Shao et al., 2024), REINFORCE (Williams, 1992), and Decoupled Clip and Dynamic Sampling Policy Optimization (DAPO) (Yu et al., 2025).

### A.1.3 ARTIFICIAL INTELLIGENCE APPLICATIONS IN CHEMISTRY

Artificial intelligence has found extensive application in the chemistry domain, using deep learning to learn from large-scale data and thereby accelerating research in complex tasks. The key areas include molecular design, property prediction, and reaction-related applications. In molecular design, the goal is to generate small molecules with desired properties while maintaining synthetic accessibility. Current approaches commonly use language models (Edwards et al., 2025), autoencoders (Hong et al., 2019), and diffusion models (Wang et al., 2025a), which together enable flexible and targeted compound generation. For molecular property prediction, the task involves forecasting properties based on molecular structure. State-of-the-art methods largely rely on pre-trained Transformer models (Song et al., 2023) and GNNs (Prakash et al., 2023).

In reaction-related tasks, forward reaction prediction aims to predict reaction outcomes from given reactants and reagents. Traditional Task-specific models for predicting organic reaction outcomes can be mainly categorized into template-based and template-free methods. Template-based methods integrate machine learning with predefined structural transformation rules, also known as reaction templates, either curated by experts or extracted from atom-mapped datasets (Chen & Jung, 2022; Sacha et al., 2021). In contrast, template-free methods employ data-driven architectures, such as graph neural networks (GNNs) or Transformer-based sequence models, to infer reaction patterns directly from large reaction corpora without relying on explicit templates (Schwaller et al., 2018; Irwin et al., 2022). LLM-based reaction predictors combine chemical text pretraining with Seq2Seq supervised fine-tuning to generate product SMILES from reactant inputs (Christofidellis et al., 2023; Zhao et al., 2024b). Recent advances further integrate reasoning modules to enhance mechanistic fidelity and prediction accuracy (Zhao et al., 2025b). Retrosynthesis planning works in reverse by deducing viable starting materials and intermediates to propose synthetic routes (Wang et al., 2022; Zhao et al., 2025a; Yao et al., 2024). Meanwhile, reaction condition recommendation seeks to suggest catalysts, solvents, and other reaction parameters for a given transformation (Gao et al., 2018; Wang et al., 2025b; Qian et al., 2023).

## A.2 Supplementary Material

Code and data at `https://github.com/GuliGuli-Boom/Reaction-Thinker`.

### A.2.1 Chain-of-Thought Dataset

Supplementary to Section **CoT Dataset for Reaction Reasoning** in the main text.

**Data Generation Script**

The data generation script and associated prompt template for Section **Stage 1: Preliminary Construction** has been made publicly available in *CoT-Gen.py* and *User-Prompt.txt*, respectively.

The prompt first provides explicit high-level instructions, guiding the model to assume the role of an expert chemist and to articulate the reaction reasoning process in a step-by-step manner.

---

**Prompt Template**

You are an expert-level AI chemist capable of explaining the reasoning process for chemical reactions step by step.

You will be given a standardized chemical reaction equation, including reactants, reagents, and products. Your task is to construct a reasoning dataset based on this equation.

For each reasoning step, provide a title describing what you did in that step and its content.

Requirements: Use 5-8 reasoning steps. Verify the given answer through different methods. Consider potential alternative answers and explain why they are rejected. Contemplate the possibility of error: If your reasoning is wrong, identify where the error could occur. Fully test all other possibilities where you might be mistaken. When you say you are rechecking, genuinely recheck using another method instead of merely stating that you are rechecking. Format your response in JSON. Each reasoning step should include the following keys:

**"Title": A description of the current reasoning step.**

**"Content": A detailed explanation of this step.**

**"Next-Action": Either "continue" or "final answer".**

---

Subsequently, a concrete reaction example is included to demonstrate the expected reasoning structure and output format, enabling the LLM to imitate both the logical progression and the structured JSON representation during data generation. The textual descriptions corresponding to the key reaction mechanisms are highlighted in **bold**.

---

**Reaction Reasoning Example**

**Reaction inputs:**

CN(C)c1ccc(-c2cnc3ccc(O[Si](C)(C)C(C)(C)C)cc3n2)cc1.[F-]

**Reaction output:**

CN(C)c1ccc(-c2cnc3ccc([OH])cc3n2)cc1

**"Title":** Analyzing the given reactants and reagents

**"Content":** The reactants and reagents include

CN(C)c1ccc(-c2cnc3ccc(O[Si](C)(C)C(C)(C)C)cc3n2)cc1, a molecule with a central aromatic ring substituted with a pyridine moiety, and [F-], a fluoride ion.

**"Next-Action":** "continue"

**"Title":** Hypothesizing the possible reaction mechanism

**"Content":** The reaction may involve the following reaction centers: The silicon atom in the silyl ether group and its attached oxygen atom; The incoming fluoride ion interacts with the silicon. The reaction describes the transformation of a silyl ether to a hydroxyl group. **This process typically involves a deprotection reaction where the silyl protecting group is removed, exposing the original alcohol.** Here, the fluorine ion acts as a nucleophile in the mechanism, which is common for desilylation reactions.

---

**"Next-Action":** continue

**"Title":** Predicting possible reactions

**"Content":** Here are the possible mechanistic steps of this reaction. Nucleophilic attack: The fluoride ion attacks the silicon atom of the silyl ether. Silicon has a high affinity for fluorine due to its ability to form strong Si-F bonds. **Cleavage of Si-O bond:** As the Si-F bond forms, the Si-O bond is weakened and subsequently cleaved. This releases the Si group (usually as a volatile SiF compound), and the hydroxyl group is restored. **Reformation of hydroxyl group:** With the removal of the silyloxy group, a hydrogen from the solvent or environment can bond to the oxygen, completing the transformation to the hydroxyl group.

**"Next-Action":** continue

**"Title":** Validating the reaction pathway

**"Content":** Fluoride ion (F-) acts as a nucleophile and attacks the silicon atom in the TBDMS group. Upon attack by fluoride, the silicon center forms a pentavalent transition state, which is relatively stable because silicon can accommodate five bonds due to its size and electronic structure. This intermediate is often depicted with a negatively charged fluoride attached to silicon, briefly forming a complex where the oxygen-silicon bond is weakened. As the intermediate collapses, **the Si-O bond between the silyl group and the oxygen atom is cleaved**, breaking the silyl ether linkage and releasing a free alcohol (-OH). After the Si-O bond has been cleaved, **the molecule is left with a free hydroxyl group (-OH) in place of the silyl ether.** The Si-F bond formation results in the formation of tert-butyldimethylsilyl fluoride (TBSF) as a byproduct. TBSF is relatively stable and does not interfere with the reaction further.

**"Next-Action":** continue

**"Title":** Considering alternative possibilities

**"Content":** The first possible reaction is that the fluoride ion could attack the electron-deficient aromatic ring near the nitrogen atoms, potentially displacing a substituent in a nucleophilic aromatic substitution. However, Nucleophilic aromatic substitution (SNAr) generally requires strong electron-withdrawing groups ortho or para to the leaving group, which are not present in this molecule. Fluoride is more likely to target the silicon atom in the silyl group due to silicon's affinity for fluoride and the relatively low activation energy required for Si-O bond cleavage. The second possible reaction is that the nucleophilic fluoride could theoretically attack the methyl groups attached to the tertiary nitrogen, leading to N-dealkylation and removing one or both methyl groups. However, N-dealkylation would typically require a more electrophilic carbon (such as a benzylic or allylic carbon) or harsher conditions, such as oxidation. In contrast, the silicon-fluoride interaction is very strong, making the silyl ether bond much more susceptible to fluoride attack than the N-C bonds in the tertiary amine. Fluoride ions could, under special conditions, participate in deprotonating an aromatic C-H bond or causing other activation of the aromatic ring. However, Aromatic C-H activation is highly unlikely under these conditions and would generally require a transition-metal catalyst and more stringent conditions. The Si-O bond in the silyl ether is far more reactive toward fluoride under mild conditions. The last possible reaction could be a fluoride-mediated deprotection of a tert-butyldimethylsilyl (TBDMS) group, restoring the free hydroxyl (-OH) functionality in the aromatic compound. The high affinity between silicon and fluoride drives the reaction forward. The formation of the strong Si-F bond makes the reaction favorable and specific for silyl deprotection. The fluoride ion selectively attacks the silicon center due to its high electrophilicity, leaving other functional groups (like the tertiary amine and aromatic rings) untouched.

**"Next-Action":** continue

**"Title":** Confirming the reasoning process

**"Content":** After reasoning and validation,

CN(C)c1ccc(-c2cnc3ccc([OH])cc3n2)cc1 is confirmed to be the final product.

**"Next-Action":** continue

**"Title":** Final reasoning result

**"Content":** Based on the above analysis, the reaction product is

CN(C)c1ccc(-c2cnc3ccc([OH])cc3n2)cc1. Instead of the silyl ether group, there is a hydroxyl (OH) group.

**"Next-Action":** final-answer

**Open-Source Samples**

A curated subset of 100 randomly selected examples is provided in *Open-Source.jsonl* for demonstration purposes. The full dataset, consisting of 119K (from USPTO-MIT) and 575K (from ORD) reaction reasoning samples, is available at our GitHub repository.

### A.2.2 TRAINING DEEPSEEK-R1-DISTILL-QWEN-7B

Supplementary to Section **Stage 2: Distillation and Validation** of **CoT Dataset for Reaction Reasoning** in the main text. We document the experimental setting and training progress of DeepSeek-R1-Distill-Qwen-7B during the CoT dataset construction process.

**Experimental Settings and Results**

The implementation details are specified in the scripts *SFT-DeepSeek-7B.sh* and *GRPO-DeepSeek-7B.sh*. The experiments are run on 8 NVIDIA A800 GPUs.

We record the reward curves during GRPO in Figure 5. We also evaluate the final models after SFT and GRPO, with detailed results presented in Table 6. This is a preliminary study to validate the integration of LLMs and RL for organic reaction prediction. This pilot study, which also yields a dataset of CoT reasoning traces, provides compelling evidence for the viability of our method. However, anticipating the limitations of a 7B model, we proceed with a more powerful DeepSeek-R1-Distill-Qwen-32B in the main experiment.

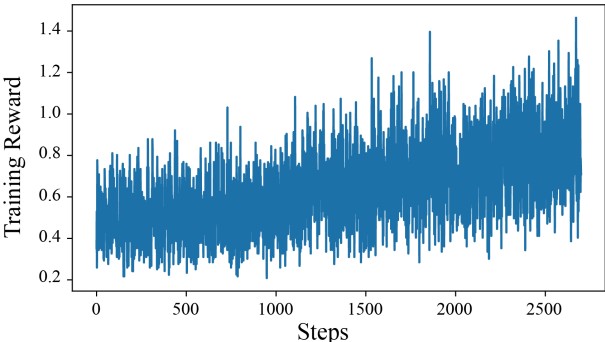

Figure 5: The reward curve during GRPO training for DeepSeek-R1-Distill-Qwen-7B.

Table 6: The Validity and Exact Match performance of DeepSeek-R1-Distill-Qwen-7B on the USPTO-MIT test set after SFT and GRPO, respectively.

|            | Validity (%) | Exact Match (%) |
|------------|--------------|-----------------|
| After SFT  | 86.4         | 22.4            |
| After GRPO | 91.9         | 35.9            |

### A.2.3 TRAINING REACTION TYPE CLASSIFIER

Supplementary to Section **Reaction Type Classifier**.

**Experimental Settings and Results**

The training and test code, as well as parameter configurations are implemented in *Classifier.py* file.

We randomly shuffled the raw Schneider-50K dataset and split it into a 40K training set and a 10K validation set. The final classifier achieves a Top-1 accuracy of 94.35% on the validation set.

### A.2.4 TRAINING RAG-BASED LLM

**Experimental Settings**

The implementation details are specified in the script *RAG-Qwen-32B.sh*. All experiments are run on 8 NVIDIA A800 GPUs.

### A.2.5 TRAINING REASONING-BASED LLM

**Experimental Settings**

The SFT implementation details are specified in *SFT-DeepSeek-32B-1.sh*, *SFT-DeepSeek-32B-2.sh*. The GRPO implementation details are specified in *GRPO-DeepSeek-32B-1.sh* and *GRPO-DeepSeek-32B-2.sh*. All experiments are run on 16 NVIDIA A800 GPUs.

## A.3 ADDITIONAL EXPERIMENTS

### A.3.1 RETROSYNTHETIC VALIDATION

This is the supplementary to Section **Analysis of Incorrect Predictions**. We conduct additional experiments for the baseline models using the new retrosynthetic validation method (in Table 7). Retro Validity assesses whether a plausible retrosynthetic route exists based on the given reaction input for products predicted by LLM. It is necessary to note that in our new experiment, Reaction-Thinker records retrosynthesis validation for both Reasoning-based and RAG-based LLMs. As a result, the score increases from 92.64% to 93.89%. These results demonstrate the superior Retro Validity of our method.

Table 7: The results for GPT-4o (marked †) are based on a randomly sampled subset of 500 instances from the ORD test set. The top results are highlighted in **bold**.

| Model | Model Type | Retro Validity (%) |
|---|---|---|
| Chemformer | Task-Specific Model | 92.91 |
| GPT-4o † |  | 31.32 † |
| DeepSeek-R1 |  | 13.89 |
| Qwen2.5-72B-Instruct |  | 1.44 |
| DeepSeek-R1-Distill-Llama-70B | General LLM | 8.22 |
| DeepSeek-R1-Distill-Qwen-32B |  | 7.83 |
| DeepSeek-R1-Distill-Qwen-14B |  | 1.98 |
| DeepSeek-R1-Distill-Qwen-7B |  | 1.90 |
| ChemDFM-13B |  | 58.60 |
| ChemDFM-8B | Chemical LLM | 57.48 |
| Text-Chem-T5 |  | 53.56 |
| **Reaction-Thinker** | **–** | **93.89** |

### A.3.2 RESULTS ON USPTO-MIT

We add the evaluation results on the USPTO-MIT test set (in Table 8). Here, we focus on evaluating task-specific models and chemical LLMs, and introduce LocalTransform (Chen & Jung, 2022) (which also demonstrated excellent performance on the USPTO-MIT dataset) as a new baseline.

It is worth noting that approximately 65% of the USPTO-MIT test samples already appear in the ORD training set. To ensure a fair comparison, we evaluated Reaction-Thinker in two settings, on the full USPTO-MIT test set, and on a filtered version (marked †) that excludes all samples seen during training.

### A.3.3 RESULTS ON RETRIEVED AND NON-RETRIEVED SUBSETS

We extend our evaluation to report baseline and ablation results on two subsets of the ORD test set, including Retrieved-case subset (reactions for which at least one similar-case was retrieved) and No-retrieval subset (reactions for which no similar-case was found in the retrieval library) (in Table 9). The results indicate that RAG-based LLM clearly benefits when similar reaction cases are retrieved, while on both subsets, the Reasoning-based LLM and other baselines show relatively small performance differences.

Table 8: Performance comparison of our method and various baseline models on the organic reaction prediction task using the USPTO-MIT test set. For task-specific models, we directly report the results (Exact Match) from their original publications.

| Model | Model Type | Validity (%) | Exact Match (%) | FTS (%) | | |
| --- | --- | --- | --- | --- | --- | --- |
| | | | | MORGAN | RDK | AVALON |
| Chemformer | | – | 90.9 | – | – | – |
| Molecular Transformer | Task-Specific Model | – | 88.6 | – | – | – |
| LocalTransform | | – | 90.8 | – | – | – |
| ChemDFM-13B | | 98.45 | 50.83 | 76.96 | 81.89 | 82.04 |
| ChemDFM-8B | Chemical LLM | 98.08 | 48.76 | 74.88 | 80.03 | 80.16 |
| Text-Chem-T5 | | 98.27 | 50.15 | 76.67 | 81.93 | 81.96 |
| **Reaction-Thinker** | – | 99.04 | 92.13 | 95.92 | 96.91 | 96.98 |
| **Reaction-Thinker** † | – | 99.02 † | 90.90 † | 95.40 † | 96.46 † | 96.51 † |

Table 9: Exact Match performance on Retrieved-case and No-retrieval subset.

| Model | Exact Match (%) Retrieved-case subset | Exact Match (%) No-retrieval subset |
| --- | --- | --- |
| Chemformer | 88.50 | 86.48 |
| DeepSeek-R1 | 11.82 | 11.05 |
| Qwen2.5-72B-Instruct | 0.55 | 0.50 |
| DeepSeek-R1-Distill-Llama-70B | 7.17 | 7.33 |
| DeepSeek-R1-Distill-Qwen-32B | 6.58 | 6.25 |
| DeepSeek-R1-Distill-Qwen-14B | 1.61 | 2.05 |
| DeepSeek-R1-Distill-Qwen-7B | 1.08 | 1.79 |
| ChemDFM-13B | 52.93 | 50.09 |
| ChemDFM-8B | 48.37 | 46.46 |
| Text-Chem-T5 | 47.91 | 47.84 |
| **Reaction-Thinker (Reasoning-based)** | 68.57 | **68.24** |
| **Reaction-Thinker (RAG-based)** | **94.70** | 31.19 |

### A.3.4 PERFORMANCE ROBUSTNESS

We conduct a thorough analysis of base models with varying parameter sizes (e.g., 7B, 8B, 14B, 32B) to systematically evaluate performance robustness (in Table 10). We perform analysis on the ORD dataset using the same training and test splits as Reaction-Thinker (32B).

The results demonstrate that our method (combining human-chemist–style reasoning and key LLM methodologies) performs robustly across various model scales, outperforms other chemical LLMs of similar size (ChemDFM-13B, ChemDFM-8B, and Text-Chem-T5), and shows consistent performance improvements as the number of model parameters increases.

Table 10: Exact Match performance of RAG-based and Reasoning-based LLMs across various model parameter sizes on the ORD dataset. The Total Accuracy is the final result obtained by weighting Qwen3 and DeepSeek-R1-Distill-Qwen at 81.7% and 18.3%, respectively.

| Model Scaling | Model | Exact Match (%) | Total Accuracy (%) |
| --- | --- | --- | --- |
| 32B | Qwen3-32B | 94.70 | 89.86 |
| | DeepSeek-R1-Distill-Qwen-32B | 68.24 | |
| 14B | Qwen3-14B | 91.08 | 85.89 |
| | DeepSeek-R1-Distill-Qwen-14B | 62.73 | |
| 8B | Qwen3-8B | 90.61 | 84.53 |
| | DeepSeek-R1-Distill-Qwen-8B | 57.39 | |
| 7B | Qwen3-7B | 89.87 | 83.69 |
| | DeepSeek-R1-Distill-Qwen-7B | 56.12 | |

### A.4 Interpretability Analysis

In this section, we try to give a detailed analysis of the interpretability advantages provided by the Reaction-Thinker. We elucidate these advantages from two main perspectives.

#### A.4.1 Human-Centered Explanations

Our method generates outputs specifically for human chemists, providing a reasoning process that aligns with their professional mindset. It delivers not just a simple answer, but a step-by-step rationale. This allows users to quickly grasp the reaction mechanism and assess the credibility of prediction by examining the correctness of the reasoning logic. Consequently, our interpretability directly serves mechanism-driven organic reaction research, going beyond prediction task.

#### A.4.2 Reliability of the Reasoning Process

It is well-known that LLM reasoning hallucinations are common. Even when the final answer is correct, the intermediate chain-of-thought may be unfaithful. We enforce CoT quality through three progressively stringent checks for this task:

- **Format Compliance:** Whether the reasoning follows a standard template.
- **Framework Conformance:** Whether it matches our predefined reasoning framework.
- **Detailed Correctness:** Whether the chain correctly tracks molecular structures, functional groups, and reaction mechanism.

In practice, we concentrate on the first two levels (format and framework) because they can be efficiently filtered using keyword and structural checks during large-scale data cleaning. As detailed in Section **CoT Dataset for Reaction Reasoning**, our data pipeline is as follows:

- **Design CoT Template:** We manually designed high-quality CoT templates. We used GPT to generate reasoning chains from selected samples, which were then reviewed and corrected by both human chemists and GPT.
- **Stage-1:** We generated numerous reasoning chains using these templates, and filtered out those violating format or framework requirements.
- **Stage-2:** We used the filtered CoT for SFT. Then we applied RL on the SFT-model, and collected correct reasoning chains as additional training data.

In the analysis of cases where the final product prediction is correct, we find that some reasoning chains contain detail errors including:

- **Incorrect descriptions of molecular structures**
- **Misnaming or mis-functionality of functional groups**
- **Mistakes in the predicted reaction mechanism**

We present a visualization of detailed analyses for two representative cases. One is the CoT Template case (in Figure 6), and the other is a case where the result is correct but the reasoning process contains the various issues listed above (in Figure 7).

Considering that verifying detailed mechanistic correctness at scale is extremely time-consuming, we test on a small sample of 50 chains. We observe that our fully trained model (SFT + RL) contained mechanistic errors in about 20% of the cases where the final product was correct. In contrast, the Stage-1 SFT-model had a significantly higher error rate of 40-60%. This indicates that our carefully constructed data and effective training methodology substantially reduce mechanistic errors while improving prediction accuracy. In other words, the explanations generated by Reaction-Thinker are highly consistent with the correct predictions, highlighting the reliability of the reasoning process.

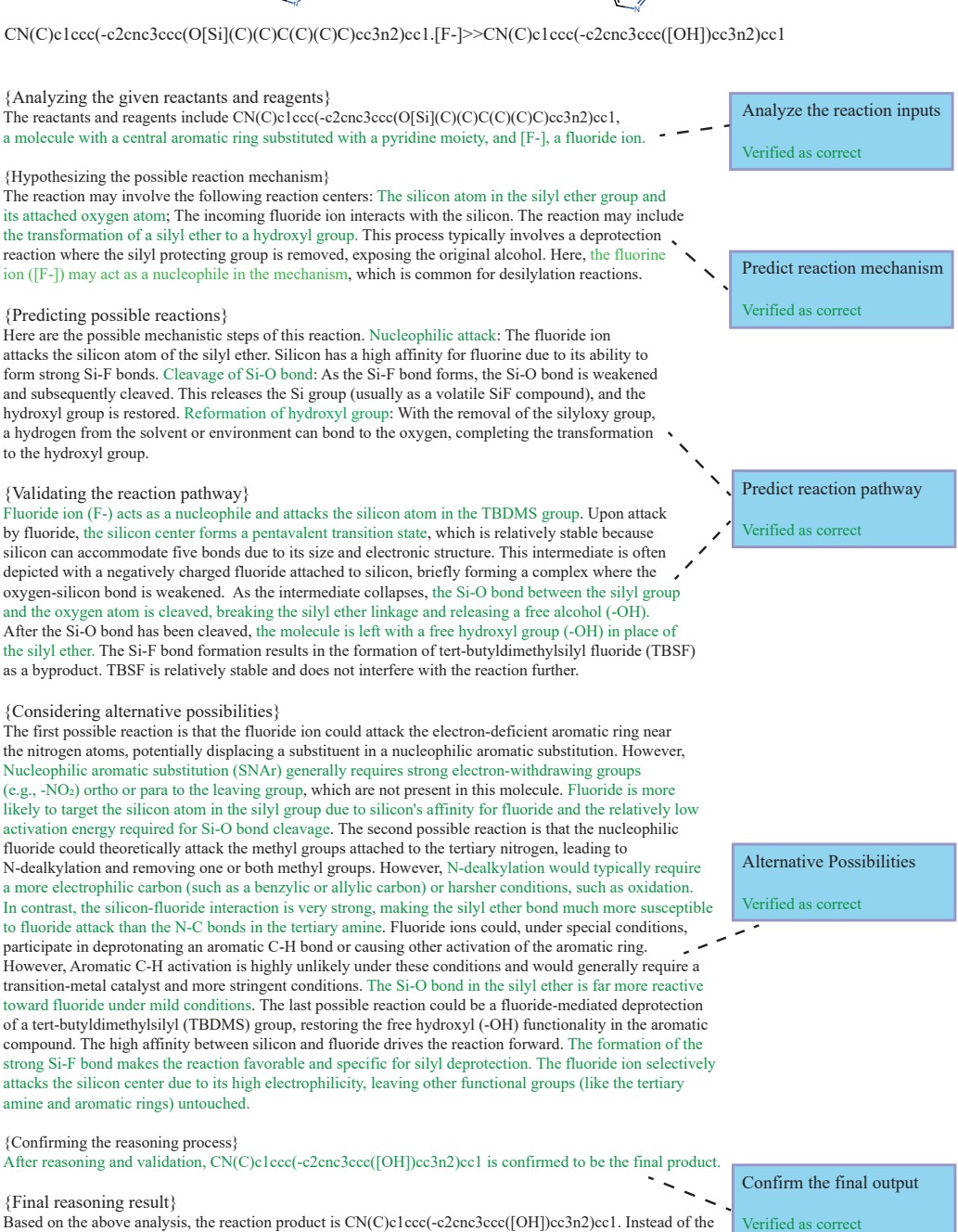

Reaction inputs          Reaction output

CN(C)c1ccc(-c2cnc3ccc(O[Si](C)(C)C(C)(C)C)cc3n2)cc1.[F-]>>CN(C)c1ccc(-c2cnc3ccc([OH])cc3n2)cc1

{Analyzing the given reactants and reagents}
The reactants and reagents include CN(C)c1ccc(-c2cnc3ccc(O[Si](C)(C)C(C)(C)C)cc3n2)cc1, a molecule with a central aromatic ring substituted with a pyridine moiety, and [F-], a fluoride ion.

Analyze the reaction inputs

Verified as correct

{Hypothesizing the possible reaction mechanism}
The reaction may involve the following reaction centers: The silicon atom in the silyl ether group and its attached oxygen atom; The incoming fluoride ion interacts with the silicon. The reaction may include the transformation of a silyl ether to a hydroxyl group. This process typically involves a deprotection reaction where the silyl protecting group is removed, exposing the original alcohol. Here, the fluorine ion ([F-]) may act as a nucleophile in the mechanism, which is common for desilylation reactions.

Predict reaction mechanism

Verified as correct

{Predicting possible reactions}
Here are the possible mechanistic steps of this reaction. Nucleophilic attack: The fluoride ion attacks the silicon atom of the silyl ether. Silicon has a high affinity for fluorine due to its ability to form strong Si-F bonds. Cleavage of Si-O bond: As the Si-F bond forms, the Si-O bond is weakened and subsequently cleaved. This releases the Si group (usually as a volatile SiF compound), and the hydroxyl group is restored. Reformation of hydroxyl group: With the removal of the silyloxy group, a hydrogen from the solvent or environment can bond to the oxygen, completing the transformation to the hydroxyl group.

Predict reaction pathway

Verified as correct

{Validating the reaction pathway}
Fluoride ion (F-) acts as a nucleophile and attacks the silicon atom in the TBDMS group. Upon attack by fluoride, the silicon center forms a pentavalent transition state, which is relatively stable because silicon can accommodate five bonds due to its size and electronic structure. This intermediate is often depicted with a negatively charged fluoride attached to silicon, briefly forming a complex where the oxygen-silicon bond is weakened. As the intermediate collapses, the Si-O bond between the silyl group and the oxygen atom is cleaved, breaking the silyl ether linkage and releasing a free alcohol (-OH). After the Si-O bond has been cleaved, the molecule is left with a free hydroxyl group (-OH) in place of the silyl ether. The Si-F bond formation results in the formation of tert-butyldimethylsilyl fluoride (TBSF) as a byproduct. TBSF is relatively stable and does not interfere with the reaction further.

{Considering alternative possibilities}
The first possible reaction is that the fluoride ion could attack the electron-deficient aromatic ring near the nitrogen atoms, potentially displacing a substituent in a nucleophilic aromatic substitution. However, Nucleophilic aromatic substitution (SNAr) generally requires strong electron-withdrawing groups (e.g., -NO₂) ortho or para to the leaving group, which are not present in this molecule. Fluoride is more likely to target the silicon atom in the silyl group due to silicon's affinity for fluoride and the relatively low activation energy required for Si-O bond cleavage. The second possible reaction is that the nucleophilic fluoride could theoretically attack the methyl groups attached to the tertiary nitrogen, leading to N-dealkylation and removing one or both methyl groups. However, N-dealkylation would typically require a more electrophilic carbon (such as a benzylic or allylic carbon) or harsher conditions, such as oxidation. In contrast, the silicon-fluoride interaction is very strong, making the silyl ether bond much more susceptible to fluoride attack than the N-C bonds in the tertiary amine. Fluoride ions could, under special conditions, participate in deprotonating an aromatic C-H bond or causing other activation of the aromatic ring. However, Aromatic C-H activation is highly unlikely under these conditions and would generally require a transition-metal catalyst and more stringent conditions. The Si-O bond in the silyl ether is far more reactive toward fluoride under mild conditions. The last possible reaction could be a fluoride-mediated deprotection of a tert-butyldimethylsilyl (TBDMS) group, restoring the free hydroxyl (-OH) functionality in the aromatic compound. The high affinity between silicon and fluoride drives the reaction forward. The formation of the strong Si-F bond makes the reaction favorable and specific for silyl deprotection. The fluoride ion selectively attacks the silicon center due to its high electrophilicity, leaving other functional groups (like the tertiary amine and aromatic rings) untouched.

Alternative Possibilities

Verified as correct

{Confirming the reasoning process}
After reasoning and validation, CN(C)c1ccc(-c2cnc3ccc([OH])cc3n2)cc1 is confirmed to be the final product.

Confirm the final output

Verified as correct

{Final reasoning result}
Based on the above analysis, the reaction product is CN(C)c1ccc(-c2cnc3ccc([OH])cc3n2)cc1. Instead of the silyl ether group, there is a hydroxyl (OH) group.

Figure 6: Detailed analyses for the CoT Template case. All critical parts are verified correct.

Reaction inputs

(1) O=C(O)C(F)(F)F
(2) COc1ccc2c(c1)N(CCCN1CCNCC1)C(=O)OC2
(3) CCN(C(C)C)C(C)C
(4) O=Cc1cc2c(cn1)OCCO2
(5) CC(=O)O[BH-](OC(C)=O)OC(C)=O
(6) [Na+]
(7) ClCCl
(8) CO

Reaction output
COc1ccc2c(c1)N(CCCN1CCN(Cc3cc4c(cn3)OCCO4)CC1)C(=O)CO2

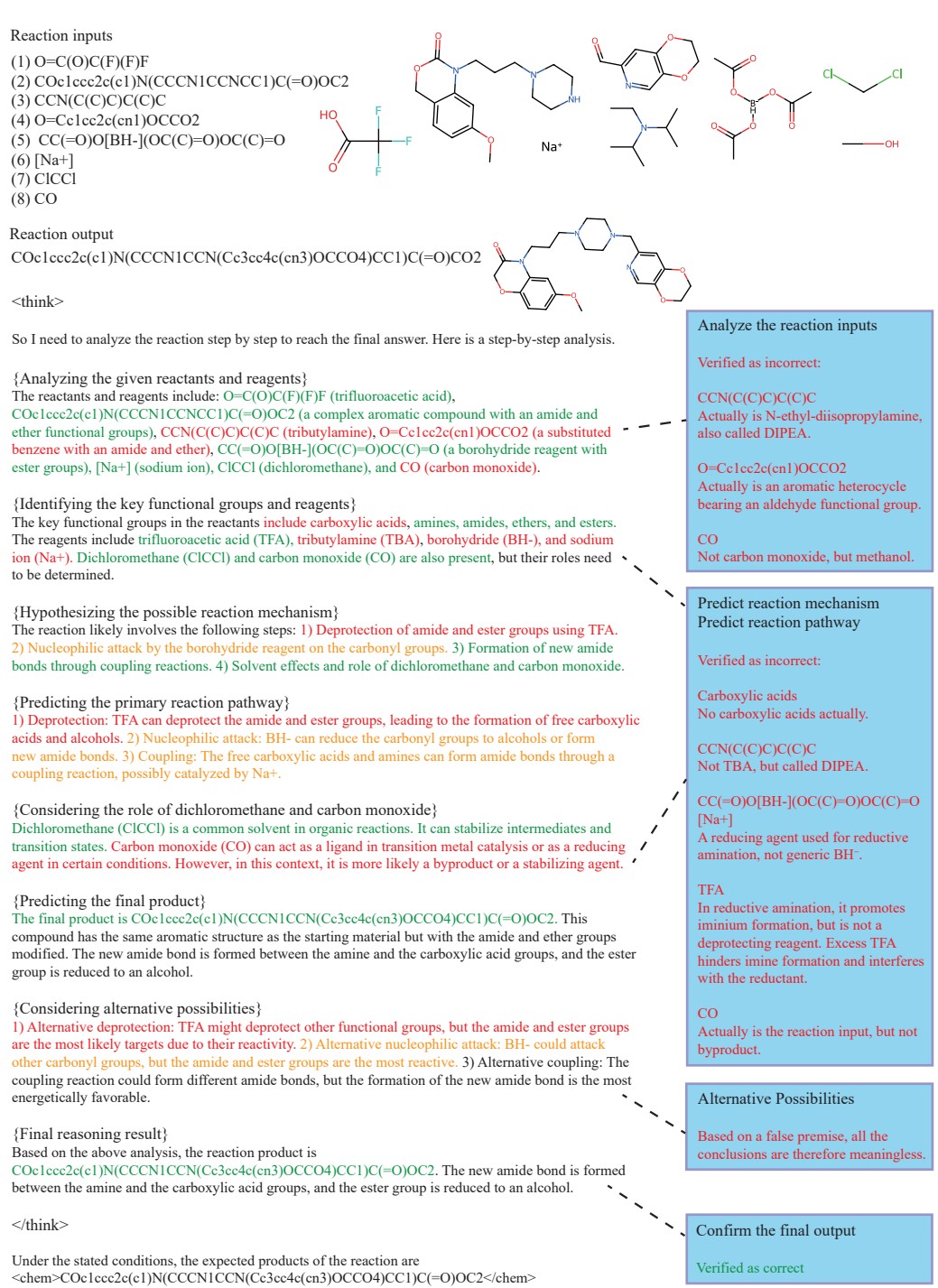

\<think>

So I need to analyze the reaction step by step to reach the final answer. Here is a step-by-step analysis.

{Analyzing the given reactants and reagents}
The reactants and reagents include: O=C(O)C(F)(F)F (trifluoroacetic acid), COc1ccc2c(c1)N(CCCN1CCNCC1)C(=O)OC2 (a complex aromatic compound with an amide and ether functional groups), CCN(C(C)C)C(C)C (tributylamine), O=Cc1cc2c(cn1)OCCO2 (a substituted benzene with an amide and ether), CC(=O)O[BH-](OC(C)=O)OC(C)=O (a borohydride reagent with ester groups), [Na+] (sodium ion), ClCCl (dichloromethane), and CO (carbon monoxide).

{Identifying the key functional groups and reagents}
The key functional groups in the reactants include carboxylic acids, amines, amides, ethers, and esters. The reagents include trifluoroacetic acid (TFA), tributylamine (TBA), borohydride (BH-), and sodium ion (Na+). Dichloromethane (ClCCl) and carbon monoxide (CO) are also present, but their roles need to be determined.

{Hypothesizing the possible reaction mechanism}
The reaction likely involves the following steps: 1) Deprotection of amide and ester groups using TFA. 2) Nucleophilic attack by the borohydride reagent on the carbonyl groups. 3) Formation of new amide bonds through coupling reactions. 4) Solvent effects and role of dichloromethane and carbon monoxide.

{Predicting the primary reaction pathway}
1) Deprotection: TFA can deprotect the amide and ester groups, leading to the formation of free carboxylic acids and alcohols. 2) Nucleophilic attack: BH- can reduce the carbonyl groups to alcohols or form new amide bonds. 3) Coupling: The free carboxylic acids and amines can form amide bonds through a coupling reaction, possibly catalyzed by Na+.

{Considering the role of dichloromethane and carbon monoxide}
Dichloromethane (ClCCl) is a common solvent in organic reactions. It can stabilize intermediates and transition states. Carbon monoxide (CO) can act as a ligand in transition metal catalysis or as a reducing agent in certain conditions. However, in this context, it is more likely a byproduct or a stabilizing agent.

{Predicting the final product}
The final product is COc1ccc2c(c1)N(CCCN1CCN(Cc3cc4c(cn3)OCCO4)CC1)C(=O)OC2. This compound has the same aromatic structure as the starting material but with the amide and ether groups modified. The new amide bond is formed between the amine and the carboxylic acid groups, and the ester group is reduced to an alcohol.

{Considering alternative possibilities}
1) Alternative deprotection: TFA might deprotect other functional groups, but the amide and ester groups are the most likely targets due to their reactivity. 2) Alternative nucleophilic attack: BH- could attack other carbonyl groups, but the amide and ester groups are the most reactive. 3) Alternative coupling: The coupling reaction could form different amide bonds, but the formation of the new amide bond is the most energetically favorable.

{Final reasoning result}
Based on the above analysis, the reaction product is COc1ccc2c(c1)N(CCCN1CCN(Cc3cc4c(cn3)OCCO4)CC1)C(=O)OC2. The new amide bond is formed between the amine and the carboxylic acid groups, and the ester group is reduced to an alcohol.

\</think>

Under the stated conditions, the expected products of the reaction are
<chem>COc1ccc2c(c1)N(CCCN1CCN(Cc3cc4c(cn3)OCCO4)CC1)C(=O)OC2</chem>

**Side annotation boxes:**

Analyze the reaction inputs

Verified as incorrect:

CCN(C(C)C)C(C)C
Actually is N-ethyl-diisopropylamine, also called DIPEA.

O=Cc1cc2c(cn1)OCCO2
Actually is an aromatic heterocycle bearing an aldehyde functional group.

CO
Not carbon monoxide, but methanol.

Predict reaction mechanism
Predict reaction pathway

Verified as incorrect:

Carboxylic acids
No carboxylic acids actually.

CCN(C(C)C)C(C)C
Not TBA, but called DIPEA.

CC(=O)O[BH-](OC(C)=O)OC(C)=O
[Na+]
A reducing agent used for reductive amination, not generic BH⁻.

TFA
In reductive amination, it promotes iminium formation, but is not a deprotecting reagent. Excess TFA hinders imine formation and interferes with the reductant.

CO
Actually is the reaction input, but not byproduct.

Alternative Possibilities

Based on a false premise, all the conclusions are therefore meaningless.

Confirm the final output

Verified as correct

Figure 7: In this case, the final product is correct, but the model exhibits substantial errors in reasoning about molecular structures, functional-group identification, and the reaction mechanisms. Content shown in green is correct, content in red is clearly erroneous, and content in yellow may be subject to debate.

