# OpenReview forum: "Towards Knowledge‑and‑Data‑Driven Organic Reaction Prediction: RAG‑Enhanced and Reasoning‑Powered Hybrid System with LLMs"
_ICLR.cc/2026/Conference — ICLR 2026 Poster_

### Official Review · Reviewer_hZJv · 2025-10-30

**Soundness:** 3
**Presentation:** 2
**Contribution:** 2
**Rating:** 2
**Confidence:** 4

**Summary:**

This work targets the forward reaction prediction problem in organic chemistry, leveraging a combination of RAG, SFT, and GRPO techniques. Compared to prior works described in the paper, Reaction-Thinker demonstrates strong performance with enhanced interpretability.

**Strengths:**

- The authors effectively combine existing methods—SFT, RAG, and GRPO—to tackle the forward reaction prediction task.
- A rigorous ablation study is presented to validate the contribution of each component in the proposed system.

**Weaknesses:**

- It is somewhat limiting that the proposed system only addresses forward reaction prediction. Given that the system is built on RAG and LLM architectures, it would be more impactful to extend it to a broader range of chemistry problems.

- Most baselines are distilled LMs from DeepSeek-R1, but comparisons with larger proprietary models such as GPT or Claude would strengthen the evaluation.

- The proposed method utilizes 32B-scale Qwen/DeepSeek models, whereas the compared chemical LLMs are mostly T5-based or smaller models (up to 13B). A more comprehensive and rigorous baseline search for forward reaction prediction is needed. The related-work section should include a dedicated discussion of existing forward-reaction models, along with direct comparisons.

- The authors use the USPTO and ORD datasets; however, to ensure a fair benchmark evaluation, it is necessary to evaluate on the USPTO/Pistachio dataset. Based solely on Table 1, it is difficult to clearly verify the claimed superiority of Reaction-Thinker.

- The paper does not provide reproducible code or pseudocode.

**Questions:**

- In Section 2.4, how exactly is the RAG-based LLM fine-tuned? Are the retrieved similar-case samples simply concatenated with the input during fine-tuning? What happens if retrieval is applied without additional fine-tuning?

- Interpretability is emphasized as one of the main motivations, but no in-depth analysis is provided. Does “interpretable” simply mean that the model outputs a reasoning trace in natural language?

- During inference, there could be cases where the final predicted product is correct but the intermediate reasoning steps are wrong. Is there any analysis of such discrepancies?

---

> ### Author Response · Authors · 2025-11-24
>
> **Response to Reviewer hZJv:**
> We sincerely thank you for your constructive feedback. Below are our concise responses point-by-point. Should you have any further questions or requests, we would be happy to provide additional replies or clarifications. We are looking forward to your further comments.
>
> **1. To Weakness-1 (About Extension to Broader Chemistry Tasks):**
>
> Thanks for your valuable suggestion. We agree that extending the system to broader chemistry problems represents an important direction. We will extend this framework to other tasks in chemistry in the future.
>
> While in this study, we deliberately focused on forward reaction prediction as our primary benchmark for the following reasons.
>
> • **Ideal Benchmark:**
> Forward prediction serves as a rigorous task that requires multi-step reasoning based on reaction mechanisms. This complexity allows us to properly **evaluate our key innovations including chemical knowledge retrieval and CoT reasoning, without introducing confounding variables from multi-task setups**.
>
> • **Focus on methods:**
> Focusing on a single task **enabled deep investigation of core components like reasoning data construction and RAG optimization**. A multi-task setup would have made our detailed ablation studies and interpretability analysis significantly less focused.
>
> • **Foundation for Expansion:**
> The framework established in our work, particularly in designing chemical reasoning chains and retrieving similar reaction cases, **provides a solid foundation for adapting this approach to other chemistry tasks** such as retrosynthesis or condition recommendation.
>
> **2. To Weakness-2 (About Comparison with Larger Proprietary Models):**
>
> Thanks for your suggestion. We have conducted additional comparisons with GPT-4o on a sampled set of 500 instances from the ORD test set. The results, now included in Table 6 of the updated manuscript, provide meaningful benchmarking against a larger proprietary model while managing evaluation costs.
>
> **Table:**  Results with GPT-4o on a sampled set from the ORD test set.
>
> | Model | Validity (%) | Exact Match (%) | Retro Validity (%) | MORGAN FTS (%) | RDK FTS (%) | AVALON FTS (%) |
> |---|:--:|:--:|:--:|:--:|:--:|:--:|
> | GPT‑4o † | 83.05 † | 28.26 † | 31.32 † | 64.93 † | 72.13 † | 72.09 † |
>
> **3. To Weakness-3 (About Limitations in Baseline Selection):**
>
> Thanks for your suggestion. We fully agree and have added a dedicated discussion of existing forward-reaction prediction models in **Section A.1.3** of the revised manuscript to better situate our work.
>
> We have also conducted an analysis of base models with varying scale to evaluate scaling effects. The results, now included in Table 9 of the updated manuscript, demonstrate that our method **performs robustly across various model scales**, outperforming Chemical LLMs of similar scale while showing consistent improvements with increased model scale.
>
> **Table:**  Performance of RAG-based and Reasoning-based LLMs across various model scale on the ORD dataset.
>
> | Scale | Model | Exact Match (%) | Total Accuracy (%) |
> |---|---|:--:|:--:|
> | 32B | Qwen3‑32B | **94.70** | **89.86** |
> | 32B | DeepSeek‑R1‑Distill‑Qwen‑32B | **68.24** | **89.86** |
> | 14B | Qwen3‑14B | 91.08 | 85.89 |
> | 14B | DeepSeek‑R1‑Distill‑Qwen‑14B | 62.73 | 85.89 |
> | 8B | Qwen3‑8B | 90.61 | 84.53 |
> | 8B | DeepSeek‑R1‑Distill‑Qwen‑8B | 57.39 | 84.53 |
> | 7B | Qwen3‑7B | 89.87 | 83.69 |
> | 7B | DeepSeek‑R1‑Distill‑Qwen‑7B | 56.12 | 83.69 |
>
> **Our comparisons are comprehensive**. We not only outperform the strongest task-specific models (e.g., Chemformer), but also surpass other chemical LLMs as well as much larger LLMs like GPT-4o. This confirms the gains stem from our method innovation, not merely model scale. The conclusion is further supported by our ablation studies across different base models.
>
> **4. To Weakness-4 (About Results on USPTO-MIT test set):**
>
> Thanks for your suggestion. The evaluation results on the USPTO-MIT test set have been added (as noted in our response to Question-3 from Reviewer 1yzM). It is worth noting that about 65% of the USPTO-MIT test samples already appear in the ORD training set. To ensure a fair comparison, we evaluated Reaction-Thinker on the full USPTO-MIT test set, and on a filtered version (marked †) that excludes all samples seen during training. These results are provided in Table 7 of the updated manuscript.
>
> **5. To Weakness-5 ( About Code Availability):**
>
> We apologize for the issue with the anonymous link. We have compressed the contents of the repository and uploaded them as **Supplementary Material**.

---

> > ### Author Response · Authors · 2025-11-24
> >
> > **1. To Question-1 (About RAG-based LLM Details):**
> >
> > Thank you for the question. During both fine-tuning and inference for our RAG-based LLM, **the user prompt sequentially includes the description of the reaction (reactants, reagents, solvents), the predicted reaction type, and the retrieved similar-case examples**. We also experimented with appending mechanistic analysis of retrieved cases in the prompt, but did not observe a performance gain.
> >
> > If we **apply RAG without SFT, the retrieved similar cases contribute very little to improving prediction performance**. The SFT step is crucial, it teaches the LLM to actually leverage the retrieved examples before generating its output.
> >
> > **2. To Question-2 (About Interpretability Analysis):**
> >
> > Thank you for this critical question regarding interpretability. We agree that its value extends far beyond outputting a natural language trace. We will address this motivation from the following two perspectives.
> >
> > • **Human-Centered Explanations:**
> > Our method generates outputs specifically for human chemists, **providing a reasoning process that aligns with their professional mindset**. It delivers not just a simple answer, but a step-by-step rationale. This allows users to quickly grasp the reaction mechanism and assess the credibility of prediction by examining the correctness of the reasoning logic. Consequently, **our interpretability directly serves mechanism-driven organic reaction research, going beyond prediction task**.
> >
> > • **Reliability of the Reasoning Process:**
> > As detailed in our response to Reviewer q5vK, we implemented a methodological pipeline to maximize the quality of the CoT. The template-based CoT data generation and rigorous data filtering established a solid foundation. Then, the tailored training approach (SFT+RL) significantly reduced the occurrence of mechanistic errors in the reasoning process from 40-60% down to 20%. This demonstrates a substantial enhancement in the reliability of the explanations generated by LLM.
> > Furthermore, the final prediction is a direct output of the reasoning chain, meaning the explanation is the genuine cause of the answer, not a post-hoc attribution. This intrinsic link ensures that **a correct prediction is highly likely to be supported by a good quality reasoning process, thereby establishing the reliability of the explanations themselves**.
> >
> > **3. To Question-3 (About Prediction-Reasoning Consistency):**
> >
> > This question is important. Although the final answer is often correct, it is very common for the intermediate chain-of-thought to be unfaithful. Existing methods can only mitigate this hallucination problem in LLM, but not fully eliminate it [6][7][8].
> >
> > In our analysis of cases where the final product prediction was correct, we found that some reasoning chains contain detail errors including:
> >
> > • **Incorrect descriptions of molecular structures**
> >
> > • **Misnaming or mis-functionality of functional groups**
> >
> > • **Mistakes in the predicted reaction mechanism**
> >
> > Verifying such detailed errors at scale is extremely challenging. For example, manually checking a single chain takes about 30 minutes. Moreover, different LLMs (e.g., GPT, Qwen, DeepSeek-R1) may disagree in their evaluation of the same chain, which makes it difficult to use their judgments for direct scoring. Therefore, **we conducted a detailed evaluation about reaction mechanisms on a small sample of 50 reasoning chains, using cross-validation among GPT, Qwen, and DeepSeek-R1**.
> >
> > Our overall finding is that in our fully trained model (SFT + RL), mechanistic errors occur in about 20% of the cases where the final product is correct. By contrast, a model fine-tuned on CoT data generated via Stage-1 (in **Section 2.3.1**) had a much higher error rate of 40-60%. This suggests that, to a large extent, **proper training methods and high-quality CoT data can reduce mechanistic mistakes while improving prediction accuracy, in the task of reaction prediction**.
> >
> > **References**
> >
> > [6] Faithful chain-of-thought reasoning. IJCNLP-AACL, 2023.
> >
> > [7] Unveiling Confirmation Bias in Chain-of-Thought Reasoning. arXiv, 2025.
> >
> > [8] Dissociation of faithful and unfaithful reasoning in llms. arXiv, 2024.

---

> ### Comment · Reviewer_hZJv · 2025-11-28
>
> Thank you for your rebuttal. While my other concerns have been resolved, I still view it as a major limitation that the model—despite its multi-stage design/multi training strategy—ultimately performs only forward reaction prediction. I hope to see future extensions toward a more versatile foundation-model framework capable of handling a broader range of chemical tasks, similar to recent general-purpose approaches. In addition, a comparison with specialized models tailored for forward reaction prediction, as well as a clearer and more comprehensive interpretability analysis, appears necessary to fully establish the model’s value.
>
> Given the improvements and clarifications, I am raising my score to 4. However, I am currently unable to modify my review on OpenReview; once the editing option becomes available, I will update my score accordingly.

---

> > ### Author Response · Authors · 2025-12-01
> >
> > We have moved the interpretability analysis into a separate section (Appendix A.4), including statistical summaries and visualizations for selected examples, so that our interpretability assessment is presented as clearly and transparently as possible.

---

> ### Author Response · Authors · 2025-11-29
>
> Thank you for your feedback and for raising your score to 4. We sincerely appreciate your time and constructive engagement.
>
> We fully understand the remaining concerns you raised, and we would like to offer the following clarifications and explanations:
>
> • **Focus on forward reaction prediction only:**
> In this work, we intentionally focus on this single task because **our primary aim is to carefully examine the feasibility and performance of combining key LLM methodologies and human-expert–style reasoning, specifically for forward reaction prediction**. We believe the insights gained here will serve as a critical foundation for future expansions.
>
> • **Comparison with specialized forward-prediction models:**
> For a fair and meaningful comparison, we have selected representative SOTA specialized models (e.g. Chemformer and Molecular Transformer) that are among the best-performing baselines for forward reaction prediction. Our method achieves superior performance on multiple key metrics compared to these baselines. **We believe these comparisons sufficiently demonstrate the advantages of our approach**.
>
> • **Interpretability analysis:**
> We fully agree that interpretability is very important. **As discussed above (in our response to Question-2, Reviewer hZJv), a full verification of every step in the chemical reasoning chain is extremely labor-intensive. Therefore, we selected a small number of examples to carry out detailed qualitative and statistical analyses**. In the revised manuscript, we will incorporate these analyses explicitly, including statistical summaries and visualizations for selected individual examples, such that our interpretability assessment is presented as clearly and transparently as possible.

---

### Official Review · Reviewer_q5vK · 2025-10-30

**Soundness:** 3
**Presentation:** 3
**Contribution:** 3
**Rating:** 6
**Confidence:** 3

**Summary:**

The paper proposes Reaction-Thinker, a hybrid “knowledge + data” framework for predicting products of organic reactions. The pipeline includes a reaction-type classifier, a similar-case retrieval library, and two predictors (RAG and CoT). The authors report SOTA Exact-Match and FTS on ORD (ORDERly split).

**Strengths:**

The method uses two routes that mirror practice in organic synthesis: cases with similar precedents are handled by RAG, while cases without precedents are handled by CoT + GRPO to strengthen step-by-step reasoning.

**Weaknesses:**

As noted in section 3.4, the Retro* check only verifies that some retrosynthetic route exists and does not enforce consistency with the given reaction conditions (precursors, catalysts etc.). Counting “retrosynthesizable” as correct may therefore overestimate forward attainability in real settings.

**Questions:**

1. The paper states that the retrieval library is built entirely from the training split. But what is the dataset scope used to construct the SFT and CoT corpus, is there any risk of data leakage?

2. Results are reported under a single fixed threshold M (RAG subset Exact-Match 94.70%, CoT subset 68.24%, overall 89.86%). How was this M chosen?

3. How is the quality of the CoT corpus ensured, and how is reaction mechanism correctness enforced? Can the method provide white-box organic reaction explanations that are consistent with the correct product prediction?

---

> ### Author Response · Authors · 2025-11-24
>
> **Response to Reviewer q5vK:**
>
> We sincerely thank you for your constructive feedback. Below are our concise responses point-by-point.
>
> **1. To Weakness-1 (About Retrosynthetic Validation):**
>
> Thank you for raising this important point. We fully agree that retrosynthetic validation, which verifies the existence of a retrosynthetic route without enforcing condition consistency, may overestimate practical forward synthetic accessibility. Therefore, **Exact Match remains our primary metric for assessing prediction accuracy, while retrosynthetic validation serves as a referenced benchmark**.
>
> Besides, to ensure a fair comparison, we also applied Retro* validation to all other baseline methods. The results can be found in our response to Question-2 from Reviewer 1yzM.

---

> ### Author Response · Authors · 2025-11-24
>
> **1. To Question-1 (About Potential Data Leakage):**
>
> Thank you for your question. The SFT corpora are derived from the training splits of Open Reaction Database. The CoT corpora are derived from the training splits of USPTO-MIT and Open Reaction Database. We have carefully filtered the USPTO-MIT training data to **ensure that no reaction SMILES from it appears in the Open Reaction Database test set, eliminating any risk of data leakage during training and inference**.
>
> **2. To Question-2 (About Parameter Selection):**
>
> This is an important question. **We carried out a series of ablation experiments over different values of the threshold M before making our final choice**. These results are now provided in Table 10 of the updated manuscript. Ultimately, we selected M=10 because it yields the highest overall accuracy.
>
> **Table:**  Exact Match performance under different RAG thresholds. Proportion means proportion with similar cases, EM means Exact Match.
>
> | M | Proportion (%) | RAG-based EM (%) | Reasoning-based EM (%) | Acc (%) |
> |---|:--:|:--:|:--:|:--:|
> | 10 | 81.70 | **94.70** | 68.24 | **89.86** |
> | 30 | 92.52 | 87.33 | **69.78** | 86.02 |
> | 40 | 94.68 | 87.99 | 68.15 | 86.93 |
> | 100 | **99.10** | 88.94 | 67.27 | 88.74 |
>
> **3. To Question-3 (About CoT Quality, Correctness, and Explainability):**
>
> Thank you for the question. It is well-known that LLM reasoning hallucinations are common, even when the final answer is correct, the intermediate chain-of-thought may be unfaithful. Existing methods can only mitigate, but not fully eliminate such problem [6][7][8].
>
> In this task, we **enforce CoT quality through three progressively stringent checks**:
>
> • **Format Compliance:**
> Whether the reasoning follows a standard template;
>
> • **Framework Conformance:**
> Whether it matches our predefined reasoning framework
>
> • **Detailed Correctness:**
> Whether the chain correctly tracks molecular structures, functional groups, and reaction mechanism.
>
> In practice, **we concentrate on the first two levels (format and framework)** because they can be efficiently filtered using keyword and structural checks during large-scale data cleaning. As detailed in **Section 2.3**, our data pipeline was as follows:
>
> • **Design CoT Templates:**
> We manually designed high-quality CoT templates. We used GPT to generate reasoning chains from selected samples, which were then reviewed and corrected by both human chemists and GPT.
>
> • **Stage-1:**
> We generated numerous reasoning chains using these templates, and filtered out those violating format or framework requirements.
>
> • **Stage-2:**
> We used the filtered CoT for SFT. Then we applied RL on the SFT-model, and collected correct reasoning chains as additional training data.
>
> We acknowledge that verifying detailed mechanistic correctness at scale remains highly challenging. Manually checking a single chain takes approximately 30 minutes. Furthermore, different LLMs (e.g., GPT, Qwen, DeepSeek-R1) can evaluate the same chain differently.
>
> However, on a small sample of 50 chains, using cross-validation among GPT, Qwen, and DeepSeek-R1, we observed that **our fully trained model (SFT+RL) contained mechanistic errors in about 20% of cases where the final product was correct**. In contrast, the Stage-1 SFT-model had a significantly higher error rate of 40-60%. This indicates that our SFT+RL training substantially reduces mechanistic mistakes while improving prediction accuracy. In other words, **the explanations generated by our final model are highly consistent with the correct prediction**.
>
> **References**
>
> [6] Faithful chain-of-thought reasoning. IJCNLP-AACL, 2023.
>
> [7] Unveiling Confirmation Bias in Chain-of-Thought Reasoning. arXiv, 2025.
>
> [8] Dissociation of faithful and unfaithful reasoning in llms. arXiv, 2024.

---

### Official Review · Reviewer_1yzM · 2025-11-05

**Soundness:** 3
**Presentation:** 3
**Contribution:** 3
**Rating:** 6
**Confidence:** 4

**Summary:**

This paper introduces Reaction-Thinker, a system for organic reaction prediction with the following pipeline: 1. a 2-layer MLP that takes molecular fingerprints and generates a reaction embedding and reaction classification; 2. a reaction embedding database for RAG; 3. a RAG LLM for reactions that have similar reactions in the database, trained with SFT; 4. a reasoning LLM trained with SFT and GRPO for reactions that don’t have similar reactions in the database.

**Strengths:**

1. *Originality:* The work presents a novel application of NLP methods (RAG, SFT/RL post-training) to chemistry tasks.

2. *Quality:* The work conducts extensive experiments and ablation studies, and the training and inference procedures are well-documented.

**Weaknesses:**

1. The methodology itself is not novel, and the novelty lies in applying existing methods to an existing task. (Although retrieval for chemistry has been previously explored as well, see, e.g., [1]).

2. The approach seems a lot more costly than, e.g., Chemformer, which may be a limitation when being applied to screen large numbers of potential reactions. (And the performance benefits w.r.t. Chemformer seem marginal.)

3. The related work section (Appendix A.1) currently only mentions related work from the NLP literature. For completeness, it should also describe past work from the AI-for-chemistry literature, including the baselines the authors cited, papers cited in the introduction section, as well as potentially other works such as [1, 2, 3].

4. The provided anonymous link to the source code doesn’t work—while I can see the directory structure of the repo, all files show up as “The requested file is not found.”

5. By the way, there’s a formatting issue where all citations appeared to have used \citet{} instead of \citep{}. There are also scattered typos (line 22 missing “a”; line 706 “LLM” -> “LLMs”; etc.).

[1] Qian, Yujie, et al. "Predictive Chemistry Augmented with Text Retrieval." *EMNLP*, 2023.

[2] Gao, Hanyu, et al. "Using machine learning to predict suitable conditions for organic reactions." *ACS Central Science 4*(11), 2018.

[3] Edwards, Carl, et al. "mCLM: A Function-Infused and Synthesis-Friendly Modular Chemical Language Model." arXiv preprint arXiv:2505.12565, 2025.

**Questions:**

1. The paper mentions the performance of RAG on reactions for which a reaction was retrieved, as well as the performance of the reasoning model on reactions for which no reaction was retrieved. I’m curious what the numbers are for the baselines and ablations for these two subsets as well.

2. For the new evaluation method with retrosynthetic validation, only the number for Reaction-Thinker was given. I’m curious what the results are for the baselines? i.e., what does an additional column in Table 1 look like for this new evaluation method?

3. For the USPTO-MIT test set, only the numbers for the reasoning model part of Reaction-Thinker (with SFT and GRPO ablations) were given (in Table 5). I’m curious what Table 1, which reports results for the ORD test set only, would look like for the USPTO-MIT test set?

While not essential, I believe that answering the above questions in the main text would strengthen the paper’s evaluation and analysis.

---

> ### Author Response · Authors · 2025-11-24
>
> **Response to Reviewer 1yzM:**
> We sincerely thank you for your constructive feedback. Below are our concise responses point-by-point.
>
> **1. To Weakness-1 (About Novelty):**
>
> Thank you for your question. It is true that RAG, SFT, and RL are well-established techniques in LLM. And **the novelty of our work lies in how to tailor and integrate existing methods for the specific task**. Similar principles have also been demonstrated in many previous works [4][5].
>
> Concretely, in the task of organic reaction prediction, our approach replicates the cognitive processes of expert chemists by referencing analogous reactions and conducting reaction mechanism-based reasoning. This dual strategy enhances both prediction performance and interpretability.
>
> Our work differs from Qian et al. [1] in two key aspects:
> • **Task focus:**
> They address condition recommendation and retrosynthesis, while we perform organic reaction prediction.
>
> • **Knowledge integration:**
> They retrieve textual descriptions relevant to a reaction, and concatenate the text embeddings with molecular representations to form a joint representation, while we retrieve SMILES-based reaction cases and directly add them into user prompt for analogical reasoning.
>
> **2. To Weakness-2 (About Computational Cost and Performance Gain):**
>
> Your concerns are important. We agree that our method is more computationally demanding than Chemformer. While Chemformer is very well suited for high-throughput screening where accuracy and efficiency are critical, **our method places a stronger emphasis on interpretability and mechanistic reasoning**.Thus, we consider our approach especially valuable for key reaction prediction or mechanism-driven research, where both accuracy and explainability matter.
>
> **3. To Weakness-3 (About Expanded Discussion of Related Work):**
>
> Thank you for pointing this out. We have added a section on AI-for-chemistry related work as Section A.1.3 in the updated paper.
>
> **4. To Weakness-4 (About Non-Functional Anonymous Code Link):**
>
> We apologize for the issue with the anonymous link. We have compressed the contents of the repository and uploaded them as **Supplementary Material**. If the problem persists, we are prepared to include the key components directly in the paper appendix.
>
> **5. To Weakness-5 (About Writing and Formatting):**
>
> We thank the reviewer for pointing out these issues. We have corrected the citation format and fix all identified typos in the updated manuscript.
>
> **References**
>
> [4] Multimodal retrieval-augmented generation for clinical use cases. Machine Learning and Knowledge Extraction, 2025.
>
> [5] HuatuoGPT-o1, Towards Medical Complex Reasoning with LLMs. CoRR, 2024.

---

> ### Author Response · Authors · 2025-11-24
>
> **1. To Question-1 (About Results on the Two Reaction Subsets):**
>
> This is an interesting point. We extend our evaluation to report baseline and ablation results on both subsets including:
>
> • **Retrieved‑case subset:** Reactions for which at least one similar-case was retrieved
>
> • **No-retrieval subset:** Reactions for which no similar-case was found in the retrieval library
>
> These results are now provided in Table 5 of the updated manuscript. They indicate that **RAG‑based LLM clearly benefits when similar reaction cases are retrieved**, while on both subsets, the Reasoning‑based LLM and other baselines show relatively small performance differences.
>
> **Table:**  Results on the two reaction subsets, EM means Exact Match.
>
> | Model | Retrieved-case subset EM (%) | No-retrieval subset EM (%) |
> |:--|:--:|:--:|
> | Chemformer | 88.50 | 86.48 |
> | DeepSeek-R1 | 11.82 | 11.05 |
> | Qwen2.5-72B-Instruct | 0.55 | 0.50 |
> | DeepSeek-R1-Distill-Llama-70B | 7.17 | 7.33 |
> | DeepSeek-R1-Distill-Qwen-32B | 6.58 | 6.25 |
> | DeepSeek-R1-Distill-Qwen-14B | 1.61 | 2.05 |
> | DeepSeek-R1-Distill-Qwen-7B | 1.08 | 1.79 |
> | ChemDFM-13B | 52.93 | 50.09 |
> | ChemDFM-8B | 48.37 | 46.46 |
> | Text-Chem-T5 | 47.91 | 47.84 |
> | **Reaction-Thinker (RAG-based)** | 68.57 | **68.24** |
> | **Reaction-Thinker (Reasoning-based)** | **94.70** | 31.19 |
>
> **2. To Question-2 (About Retrosynthetic Validation):**
>
> This is a valuable suggestion. We have conducted the additional experiments for the baseline models using the new retrosynthetic validation method. These results are now provided in Table 6 of the updated manuscript.
>
> **Table:**  Results of retrosynthetic validation.
>
> | Model | Retro Validity (%) |
> | :--- | :--: |
> | Chemformer | 92.91 |
> | Molecular Transformer | -- |
> | GPT-4o † | 31.32 † |
> | DeepSeek-R1 | 13.89 |
> | Qwen2.5-72B-Instruct | 1.44 |
> | DeepSeek-R1-Distill-Llama-70B | 8.22 |
> | DeepSeek-R1-Distill-Qwen-32B | 7.83 |
> | DeepSeek-R1-Distill-Qwen-14B | 1.98 |
> | DeepSeek-R1-Distill-Qwen-7B | 1.90 |
> | ChemDFM-13B | 58.60 |
> | ChemDFM-8B | 57.48 |
> | Text-Chem-T5 | 53.56 |
> | **Reaction-Thinker** | **93.89** |
>
> **3. To Question-3 (About Results on USPTO-MIT test set):**
>
> We have now added the evaluation results on the USPTO-MIT test set. It is worth noting that approximately 65% of the USPTO-MIT test samples already appear in the ORD training set. To ensure a fair comparison, we evaluated Reaction-Thinker in two settings, on the full USPTO-MIT test set, and on a filtered version that excludes all samples seen during training (marked †). These results are now provided in Table 7 of the updated manuscript.
>
> **Table:**  Results on USPTO-MIT test set, EM means Exact Match.
>
> | Model | Model Type | Validity (%) | EM (%) | | FTS (%)  ||
> |---|:--:|:--:|:--:|:--:|:--:|:--:|
> | | | | | MORGAN | RDK | AVALON |
> | Chemformer | Task‑Specific Model | -- | 90.9 | -- | -- | -- |
> | Molecular Transformer | Task‑Specific Model | -- | 88.6 | -- | -- | -- |
> | LocalTransform | Task‑Specific Model | -- | 90.8 | -- | -- | -- |
> | ChemDFM‑13B | Chemical LLM | 98.45 | 50.83 | 76.96 | 81.89 | 82.04 |
> | ChemDFM‑8B | Chemical LLM | 98.08 | 48.76 | 74.88 | 80.03 | 80.16 |
> | Text‑Chem‑T5 | Chemical LLM | 98.27 | 50.15 | 76.67 | 81.93 | 81.96 |
> | **Reaction‑Thinker** | -- | 99.04 | 92.13 | 95.92 | 96.91 | 96.98 |
> | **Reaction‑Thinker †** | -- | 99.02 † | 90.90 † | 95.40 † | 96.46 † | 96.51 † |

---

### Author Response · Authors · 2025-11-27

We sincerely thank the reviewers for their constructive and insightful comments on our submission. We have carefully considered all suggestions, and in response we have revised the manuscript accordingly, including adding new experiments, improving clarity, and addressing all raised concerns. We hope that our revisions have strengthened the work, and respectfully invite you to examine our response. Should you have any further questions or requests, we would be happy to provide additional replies or clarifications.

---

### Author Response · Authors · 2025-12-01

The latest version of the manuscript incorporates all the requested changes: additional experiments have been added in Appendix A.3 for ease of reference, and all necessary revisions in response to the reviewer comments have been applied throughout the paper.

---

### Meta-Review · Area_Chair_pbLm · 2026-01-02

**Summary:**

This paper introduces Reaction-Thinker, a hybrid, knowledge‑and-data‑driven system that is enhanced by Retrieval‑Augmented Generation (RAG) and powered by advanced reasoning, improving both the interpretability of prediction process and the explainability of results.

Reviewers highlighted (i) the effective integration of established LLM techniques (RAG, SFT, GRPO/RL) for reaction prediction, (ii) strong empirical performance supported by rigorous ablations, and (iii) generally clear documentation of training/inference procedures.

In rebuttal and revision, the authors substantively addressed key concerns by expanding AI-for-chemistry related work and fixing reproducibility via releasing code/data in the supplementary materials, adding targeted evaluations (e.g., retrieved vs. no-retrieval subsets, retrosynthetic validation for baselines, and USPTO-MIT results), and clarifying methodology choices such as the RAG threshold selection (M=10) and data-leakage mitigation.

While some limitations remain (e.g., limited scope to forward prediction, and modest methodological novelty / computational cost concerns), the paper provides a well-motivated, carefully validated model  with strong results and improved transparency; I recommend acceptance.

**Reviewer Concerns:**

See summary.

**Reviewer Scores:**

Across the three reviews, two reviewers rated the paper 6 (“marginally above the acceptance threshold”), while the third reviewer initially rated it 2 (“reject”) but later stated they were raising their score to 4 after the rebuttal.

---

### Decision · Program_Chairs · 2026-01-26

Accept (Poster)